# Vision2Web: A Hierarchical Benchmark for Visual Website Development with Agent Verification

Zehai He [1] [*]   Wenyi Hong [1] [*]   Zhen Yang [1]   Ziyang Pan [2]   Mingdao Liu [1]   Xiaotao Gu [2]   Jie Tang [1]

## Abstract

Recent advances in large language models have improved the capabilities of coding agents, yet systematic evaluation of complex, end-to-end website development remains limited. To address this gap, we introduce Vision2Web, a hierarchical benchmark for visual website development, spanning from static UI-to-code generation, interactive multi-page frontend reproduction, to long-horizon full-stack website development. The benchmark is constructed from real-world websites and comprises a total of 193 tasks across 16 categories, with 918 prototype images and 1,255 test cases. To support flexible, thorough and reliable evaluation, we propose a workflow-based agent verification paradigm based on two complementary components: a GUI agent verifier and a VLM-based judge. We evaluate multiple visual language models instantiated under different coding-agent frameworks, revealing substantial performance gaps at all task levels, with state-of-the-art models still struggling on full-stack development. Project page: https://vision2web-bench.github.io/.

## 1. Introduction

The reasoning and coding capabilities of autonomous software agents have been significantly advanced through the development of large language models (LLMs) (Dong et al., 2025). Equipped with these models, contemporary coding agents are capable of performing end-to-end software development tasks encompassing system design, data processing, code generation, and project debugging.

Despite these advances, existing benchmarks for coding agents remain fundamentally limited in scope and rigor:

- **Limited task formulation.** Prominent benchmarks such as SWE Bench and its variants (Jimenez et al., 2024; Yang et al., 2024) focus on incremental, issue-driven code edits, capturing localized development skills but failing to evaluate holistic, end-to-end software engineering capabilities.
- **Misaligned multimodal coverage.** While recent text-only benchmarks, including VIBE Bench (MiniMax, 2025) and WebGen Bench (Lu et al., 2025), have begun exploring end-to-end development scenarios, multimodal benchmarks remain largely restricted to static webpage reproduction tasks such as Design2Code (Si et al., 2025).
- **Insufficient verification mechanisms.** Despite initial attempts toward end-to-end development (Lu et al., 2025), reliably and reproducibly assessing complex interactions and long-horizon system outcomes remains challenging, due to underspecified task definitions and insufficiently constrained verification procedures.

To address these gaps, we introduce **Vision2Web**, a hierarchical benchmark that enables autonomous evaluation of multimodal coding agents on visual website development via agent verification. *As a task formulation, website development naturally satisfies these requirements:* it spans the full software lifecycle and requires coordinated understanding of visual prototypes, textual requirements, and codebases, making it an ideal testbed for evaluating long-horizon multimodal agent intelligence. Overall, Vision2Web is designed around three core principles:

**Capability disentanglement.** To enable explicit failure attribution across development stages, Vision2Web organizes tasks into three progressively harder levels—*static webpage generation*, *interactive frontend development*, and *full-stack website construction*—with each level building upon the previous one, enabling systematic diagnosis of agent capabilities from fine-grained visual understanding to holistic system construction.

**Verifiable task construction.** Rather than relying on underspecified synthetic tasks, Vision2Web is curated from publicly accessible websites through a rigorous multi-stage pipeline that integrates large-scale data collection, auto-

---

Work was done when ZH, WH, ZY, ML interned at Zhipu AI. [*]Equal contribution [1]Tsinghua University [2]Zhipu AI. Correspondence to: Jie Tang <jietang@tsinghua.edu.cn>.

*Proceedings of the 43rd International Conference on Machine Learning*, Seoul, South Korea. PMLR 306, 2026. Copyright 2026 by the author(s).

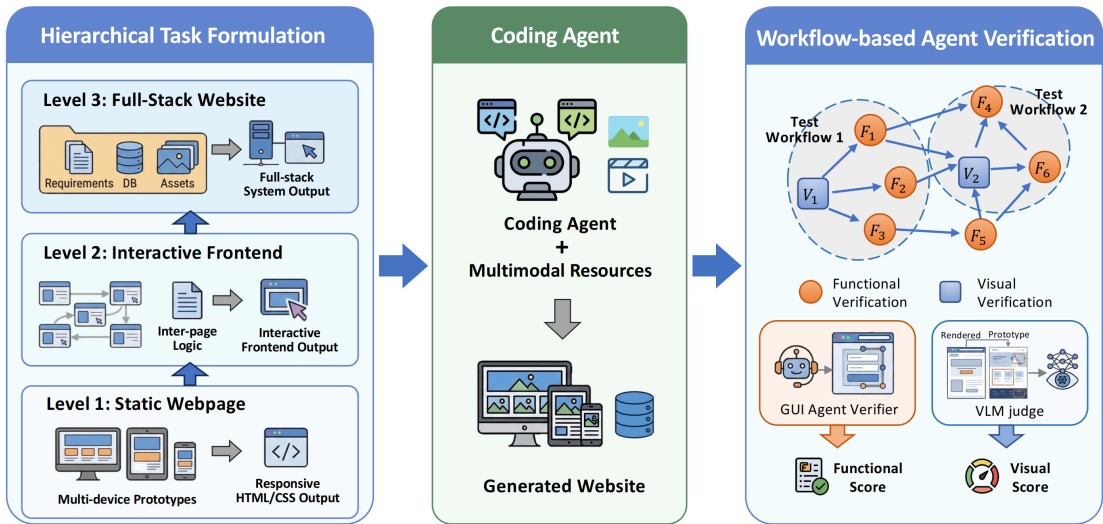

*Figure 1.* Overview of Vision2Web, a hierarchical benchmark for visual website development. Tasks span three levels—static webpages, interactive frontends, and full-stack websites—requiring agents to integrate visual prototypes with textual specifications. Evaluation is performed via a workflow-based agent verification paradigm, measuring functional correctness and visual fidelity.

*Table 1.* Comparison of Existing Benchmarks for Software Engineering and Web Development Tasks.

| Benchmark | # Tasks | # Test Cases | # Prototypes | # Multimodal | Task Types |
|---|---|---|---|---|---|
| SWE-Bench Multimodal | 617 | – | – | ✓ | Issue fixing |
| Design2Code | 484 | – | 484 | ✓ | Single page UI-to-Code |
| WebGenBench | 101 | 647 | – | ✗ | Website generation |
| Vision2Web | 193 | 1255 | 918 | ✓ | Full-stack website development |

mated filtering and agent-assisted annotation. The resulting benchmark comprises 193 website development tasks with 1255 test cases spanning four major website categories and 16 subcategories, closely reflecting the diversity of real-world websites.

**Reliable automated evaluation.** Automated assessment of end-to-end software systems remains challenging due to implementation diversity. Accordingly, Vision2Web adopts a workflow-based agent verification paradigm in which GUI agents execute expert-designed test workflows encoding multi-step, interdependent functionalities, while a dedicated VLM-based judge quantitatively evaluates visual fidelity against UI prototypes. This coordinated design enables reproducible and objective evaluation without sacrificing the flexibility of agent-based interactions.

Our experiments reveal notable gaps in the capabilities of state-of-the-art coding agents across all three levels, highlighting limitations in cross-modal reasoning, long-horizon task planning, and multi-page coordination. These insights provide a foundation for future research in advancing agent reasoning and software development performance.

In summary, Vision2Web offers the following three major contributions:

- **Hierarchical Task Design:** A hierarchical task formulation that systematically disentangles agent capabilities across stages of visual website development.
- **Realistic Multimodal Data:** A large-scale benchmark grounded in real-world websites with explicit specifications, enabling evaluation under realistic multimodal constraints.
- **Workflow-Based Agent Verification:** A reproducible, implementation-agnostic evaluation paradigm that combines structured workflows with agent execution to assess both functional correctness and visual fidelity in end-to-end website development.

**Conflict of Interest Disclosure.** Some authors were affiliated with or interned at Zhipu AI during this work. The GUI agent verifier used in our evaluation is instantiated with GLM-4.6V, a model developed by Zhipu AI. We disclose this relationship for transparency.

## 2. Overview of Vision2Web

This section provides an overview of Vision2Web, including its task formulation dataset construction and overall statistics.

## 2.1. Task Formulation

Visual website development encompasses a range of capabilities, from interpreting UI prototypes to managing interaction-driven application states and page transitions, ultimately delivering full-stack system delivery. To systematically evaluate these competencies, we formalize the coding task via a three-level hierarchical framework, with each level targeting a distinct set of critical skills.

**Level 1: Static Webpage.** This level evaluates models' ability to interpret UIs and generate executable code in a *device-responsive* setting. Each task provides prototype images of the same webpage across desktop, tablet, and mobile, with resolution specifications. Models must produce a single static webpage that faithfully reproduces layout, visual content, and styling at each resolution.

**Level 2: Interactive Frontend.** At this level, inputs include multiple prototype images and text describing inter-page logical relationships. Models must generate a fully interactive multi-page frontend that preserves structural consistency and coherent navigation flows, assessing the ability to reason across pages and organize components in a multimodal context.

**Level 3: Full-Stack Website.** This level simulates realistic engineering scenarios, providing structured requirement documents alongside prototype images. Agents are expected to interpret requirements, manage complex application states, perform integrated debugging, and deliver cohesive full-stack systems, evaluating comprehensive end-to-end software engineering capabilities.

All tasks include a multimedia resource library with images, icons, videos and fonts to simulate realistic development. Moreover, each task is defined with specific and unambiguous requirements. This hierarchical design enables Vision2Web to systematically assess model capabilities across all stages of visual website development.

## 2.2. Dataset Construction

To ensure high-quality, contamination-free evaluation data, Vision2Web is constructed through a multi-stage pipeline that refines large-scale web corpora into realistic, well-defined tasks suitable for systematic evaluation. All test tasks are sourced exclusively from the C4 (Raffel et al., 2020) validation set to avoid potential leakage from popular websites, and we apply the following three-stage filtering pipeline to guarantee dataset quality and diversity:

**Structural Assessment.** Following principles adapted from the Design2Code (Si et al., 2025) benchmark, DOM-level properties, including HTML tag distribution, DOM tree depth, and token length are analyzed. Pages with overly simple layouts, malformed structures, or insufficient se-

mantics are excluded, reducing the candidate set to 63,515 websites.

**Content Screening.** In the next stage, candidate websites are filtered for content and design quality using VLM-based scoring, retaining only 7,391 pages that demonstrate functional richness, modular clarity, and visual coherence. Pages lacking meaningful interactive components, exhibiting poor layout organization, or offering limited functional coverage are excluded.

**Manual Review.** Remaining websites undergo manual review by annotators across all task levels. Reviewers evaluate each website based on multiple criteria, including page consistency and quality across device resolutions, implementation difficulty, overall page dimensions, and the clarity and richness of interactive functionality. Websites are also selected to ensure balanced coverage across all categories, preserving diversity in content, layout, and interaction patterns.

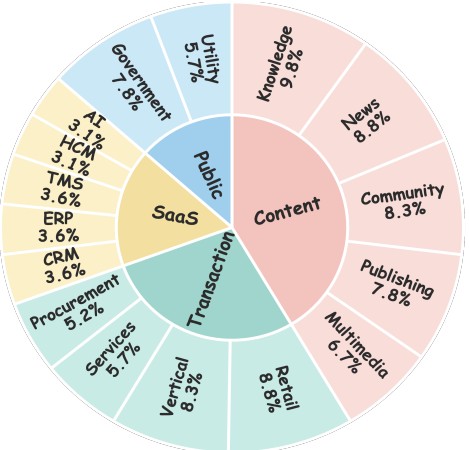

*Figure 2.* Task distribution of Vision2Web across four major categories and 16 subcategories.

*Table 2.* Average prototype images, test cases, and input text tokens for Vision2Web across three task levels.

| Level | Webpage | Frontend | Website |
|---|---|---|---|
| Avg Prototype Images | $3 \pm 0$ | $5.9 \pm 0.5$ | $8.5 \pm 2.3$ |
| Avg Test Cases | − | $7.5 \pm 4.5$ | $28.2 \pm 12.0$ |
| Avg Text Tokens($10^3$) | − | $1.0 \pm 0.6$ | $4.3 \pm 1.1$ |

## 2.3. Dataset Statistics

Overall, Vision2Web provides a hierarchical benchmark for evaluating the visual website development capabilities of multimodal coding agents. The benchmark is extensive and well-structured, comprising a total of 21516 input files, in-

cluding 918 prototype images and 1255 test cases, ensuring a rich and diverse evaluation set. It spans 193 tasks across three levels of increasing complexity: 100 static webpage tasks, 66 interactive frontend tasks, and 27 full-stack website tasks. To ensure representativeness, the tasks in Vision2Web are drawn from websites across four major categories which are further divided into 16 subcategories, with an overall distribution that closely reflects the diversity of real-world websites.

Table 2 summarizes the dataset statistics of Vision2Web. Task complexity increases from static webpages to interactive frontends and full-stack websites, reflected in the number of prototype images, test cases, and text tokens. Static webpages focus on visual fidelity, frontends add navigation interactions, and full-stack websites combine extensive content with complex functionality.

## 3. Workflow-Based Agent Verification

End-to-end website evaluation presents significant challenges for both functional and visual testing. In functional testing, traditional unit tests are often infeasible for diverse software implementations. Although autonomous LLM- or VLM-based agents have been explored as flexible evaluators more recently, they frequently exhibit unconstrained execution when required to handle diverse website realizations and provided with loosely specified objectives (e.g., "test the login function"), leading to unstable behaviors and poor reproducibility. Meanwhile, visual testing faces analogous constraints. Traditional engineering-oriented UI tests, such as rule-based scripts and handcrafted assertions, are brittle to layout changes and implementation differences. Pixel-level comparisons, while effective for static renderings, are limited to snapshots and rely on low-level appearance similarity that often diverges from human perceptual judgments.

To address these challenges, Vision2Web adopts a *workflow-based agent verification* paradigm. The core idea is to preserve the *flexibility* of agent-based interaction and its *alignment* with human visual preferences, while constraining execution through structured test workflows and explicitly defined verification nodes to achieve *reproducibility*. This design enables implementation-agnostic evaluation with controlled variance, allowing both functional correctness and visual fidelity to be assessed within a unified framework.

### 3.1. Overall Test Workflow Design

A single end-to-end website evaluation typically involves multiple user interactions and verification steps. A representative testing procedure may include authentication, multi-page navigation, followed by functional and visual verification, all of which are usually performed sequentially by human testers. These steps are inherently interdependent: later verifications rely on the successful execution of earlier

interactions and operate over shared application states.

Building on this observation, Vision2Web formalizes end-to-end testing as a *directed dependency graph*, where each node represents a self-contained verification sub-procedure (e.g., functional or visual) and edges encode sequential dependencies and shared states. Each node comprises a sequence of interactions that brings the application into a target state, followed by verification. This abstraction explicitly captures dependency structures, enabling more structured and reproducible automated evaluation.

Under this abstraction, autonomous evaluation is realized by instantiating the graph into a set of agent-executable subgraphs, each corresponding to a coherent interaction trajectory under a shared application context, referred to as **test workflows**. The collection of workflows jointly covers all nodes in the graph.

In Vision2Web, test workflows are constructed following two principled guidelines that balance evaluation stability and coverage efficiency:

- **Decoupling dependent test nodes.** Test cases that span multiple functional modules (e.g., product browsing, shopping cart, and checkout) are separated into distinct workflows. This design mitigates error accumulation and propagation along excessively long interaction chains, ensuring that failures in earlier steps do not obscure the evaluation of later components.
- **Integrating related test nodes.** Test cases that operate within the same application context, such as multiple UI interactions within a single page or functional module, are grouped into a single workflow. This reduces redundant setup and navigation while enabling coherent verification under a shared interface context.

### 3.2. Design of Verification Nodes

Each verification node in the test workflows corresponds to a verification sub-procedure targeting a specific aspect of website correctness. Vision2Web explicitly categorizes verification nodes into two complementary types: **functional verification nodes** and **visual verification nodes**, each with a dedicated verifier tailored to its characteristics. In practice, verifiers use autonomous agents guided by expert-designed workflows and structured node specifications, constraining *how* agents act while preserving flexibility in *what* is verified, enabling reliable and systematic assessment across diverse website implementations.

**Functional Verification Nodes (GUI Agent Verifier).** Functional verification nodes assess functional correctness and interaction fidelity, reported as the Functional Score (FS) in Vision2Web. Each functional verification node is formalized as a 3-tuple $n_i = \langle O_i, A_i, V_i \rangle$, where $O_i$ spec-

ifies the testing objective, $A_i$ defines guided actions that constrain the agent's interactions, and $V_i$ encodes validation criteria such as logical assertions or state-based checks. Unlike conventional test objectives that only specify desired outcomes, the explicit modeling of $A_i$ prevents agents from exploring unnecessary modules or exploiting unintended actions, thereby improving evaluation reproducibility.

Vision2Web employs a GUI agent as the functional verifier to flexibly handle diverse website implementations. Specifically, we instantiate the verifier using the task execution protocol of WebVoyager (He et al., 2024), although the framework itself is agnostic to the specific agent architecture. At each functional verification node $n_i$, the agent is provided with an explicitly constructed context: $C_i = \{\mathcal{H}_{<i}, O_i, A_i, V_i\}$, where $\mathcal{H}_{<i}$ records the objectives and actions from all preceding verification nodes. By exposing both historical context and node-specific guidance, the agent can reason about temporal dependencies and state transitions in a controlled and reproducible manner. And the overall Functional Score (**FS**) for a task level in Vision2Web is computed as the proportion of passed functional verification nodes.

---

**Algorithm 1** Workflow-Based Agent Verification

---

**input** Workflow $\mathcal{W} = (n_1 \rightarrow \cdots \rightarrow n_t)$, initial state $S_0$
**output** Aggregate functional and visual scores $(\mathcal{F}, \mathcal{V})$
  $\mathcal{H}, \mathcal{F}, \mathcal{V} \leftarrow \emptyset$
  **for** $n_i \in \mathcal{W}$ **do**
    **if** $n_i$ is Functional verification **then**
      $(F_i, S_{i+1}) \leftarrow$ GUIAgentVerifier$(\mathcal{H}, O_i, A_i, V_i, S_i)$
      $\mathcal{F} \leftarrow \mathcal{F} \cup \{F_i\}; \mathcal{H} \leftarrow \mathcal{H} \cup \{(O_i, A_i)\}$
    **else if** $n_i$ is Visual verification **then**
      $(V_i, S_{i+1}) \leftarrow$ VLMBasedJudge$(P_i, S_i)$
      $\mathcal{V} \leftarrow \mathcal{V} \cup \{V_i\}$
    **end if**
  **end for**
  **return** $(\mathcal{F}, \mathcal{V})$

---

**Visual Verification Nodes (VLM Judge).** Visual verification nodes assess visual fidelity by comparing rendered pages against reference prototypes, reported as the Visual Score (VS). Each visual verification node is formalized as $n_i = \langle P_i \rangle$, where $P_i$ denotes the target prototype.

Upon reaching a visual verification node, a dedicated VLM judge is invoked to assess visual consistency between the rendered page and the prototype. The judge performs component-level comparisons, assigning fidelity scores to corresponding functional blocks according to predefined visual rubrics. The overall visual score is computed as the average of all block-level scores, providing a straightforward measure of visual consistency. The scoring is implemented via a finely designed, structured prompt that ensures consistent component-level evaluation. For a given task level, the

Visual Score (**VS**) is calculated as the average score across all prototypes. Details of the GUI agent configuration and the complete prompts are provided in Appendix A.2.2.

### 3.3. Agent-Assisted Annotation

The workflow abstraction also provides a structured basis for test case annotation. In Vision2Web, annotation is performed through collaboration between experienced PhD researchers and Claude Code (Anthropic, 2026a), with the annotation strategy adapted to task complexity. Static webpage tasks require only resolution-specific visual checks, interactive frontend tasks focus on navigation and UI-state consistency across prototypes, and full-stack websites require dedicated workflows for long-horizon dependencies, cross-module interactions, persistent states, and boundary cases. Since purely agent-driven annotation may omit key dependencies or requirement-critical edge cases, we adopt a three-stage expert-in-the-loop pipeline.

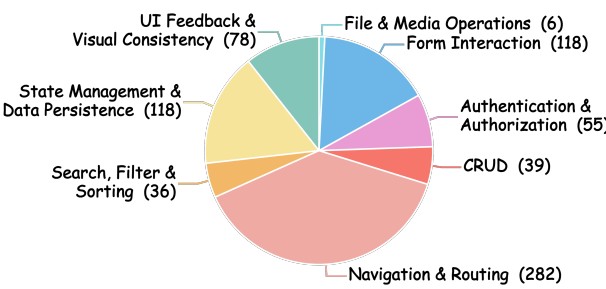

*Figure 3.* Distribution of test cases across website-level tasks in Vision2Web.

**Stage 1: Expert Workflow Drafting.** Domain experts first draft high-level workflows from requirement documents and design prototypes, specifying user goals, prerequisite states, interaction sequences, expected outcomes, and shared application states. The drafts follow two principles: decoupling dependent test nodes to reduce error propagation, and integrating related validations to avoid redundant setup and navigation.

**Stage 2: Agent-Assisted Workflow Refinement.** Based on the expert drafts, Claude Code translates high-level workflows into executable objective–action–validation structures, clarifies interaction steps and validation criteria, and helps identify missing boundary cases. The refinement is constrained by the original requirements and expert drafts, so the agent serves as an annotation assistant rather than an independent source of ground truth.

**Stage 3: Human Quality Control.** Human reviewers examine the refined workflows for completeness, correctness, and

consistency with the requirement documents and design prototypes. They remove redundant or ambiguous checks and ensure that validation criteria are implementation-agnostic and based on observable user-facing behavior.

Figure 3 illustrates the distribution of test case types across full-stack tasks, highlighting the resulting diversity and coverage of functional scenarios.

## 4. Experiments

Employing Vision2Web, we evaluate state-of-the-art multimodal models across coding agent frameworks and task levels to reveal their limitations in visual website development.

### 4.1. Settings

We evaluate eight state-of-the-art multimodal models, including Claude-Opus-4.5, Claude-Sonnet-4.5 (Anthropic, 2025), Gemini-3-Pro-Preview, Gemini-3-Flash-Preview (Google, 2026a), GPT-5 (OpenAI, 2025a), Seed-1.8-VL (ByteDance Seed, 2026), and Qwen3-VL-32B/8B-Instruct (Bai et al., 2025), integrated into two coding agent frameworks: OpenHands (Wang et al., 2025) and Claude Code (Anthropic, 2026a). Evaluations are conducted in a containerized environment preconfigured with frontend, backend, and database dependencies. For each task, all inputs—including prototype images, textual requirements, and multimedia resources—are provided in the working directory along with carefully designed prompts guiding the required level of project completion (see Appendix A.3.1 for details). To standardize deployment, each agent generates a startup script to run projects on a fixed port, with up to three iterations allowed to collect more analyzable evaluation results. Deployments exceeding 10 minutes or producing errors are treated as failures. For evaluation, the GUI agent verifier is instantiated with GLM-4.6V (GLM-V Team et al., 2025), while the VLM-based judge uses Gemini-3-Pro-Preview.

### 4.2. Main Results

The results of Vision2Web are shown in Table 3. Through a detailed, fine-grained analysis of evaluation results across task levels, models, and device settings, we derive the following key findings.

**Finding 1:** *Agent performance degrades consistently as task complexity increases across the three task levels.*
Table 3 shows that as tasks become more complex, all agents exhibit noticeable performance drops. Under the OpenHands framework, Gemini-3-Pro-Preview achieves the strongest performance on *static webpages*, with average scores of 63.3 on desktop layouts, 55.8 on tablet, and 48.3 on mobile, alongside a deployment success rate of 95%.

However, on *full-stack* tasks, its performance drops sharply to a Visual Score (VS) of 11.7, a Functional Score (FS) of 22.6, and a Deployment Success Rate (DSR) of 77.8%, highlighting the progressive difficulty across hierarchical task levels. Claude-Opus-4.5 maintains relatively strong performance across levels, but still exhibits measurable declines in both functional correctness and visual fidelity on full-stack tasks, indicating inherent limitations even for top-performing agents.

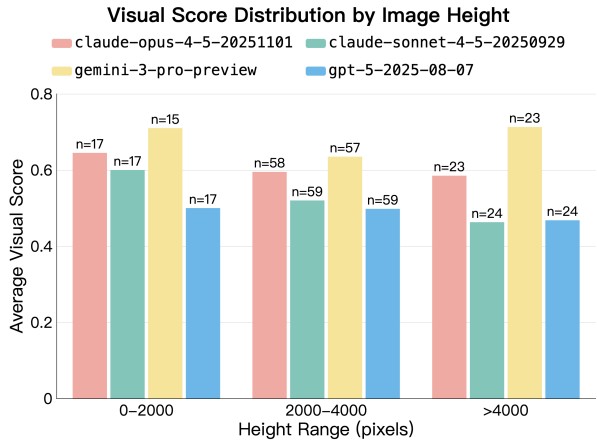

*Figure 4.* Distribution of Visual Scores (VS) across prototype heights for representative models under the OpenHands framework.

**Finding 2:** *Agent performance systematically degrades when adapting to smaller device form factors and more visually complex prototype images.*
Notably, static webpage tasks reveal consistent device-dependent drops: desktop interfaces achieve the highest fidelity, while tablet and mobile layouts show 10–20% lower scores even for top agents such as Gemini-3-Pro-Preview and Claude-Opus-4.5. Figure 4 further shows that larger, denser prototype images induce additional performance declines, reflecting current agents' limited capacity to process and reason over complex visual inputs.

**Finding 3:** *Across all evaluated agents, Claude-Opus-4.5 consistently achieves the strongest performance, clearly outperforming alternatives on complex tasks.*
Across both the Claude Code and OpenHands frameworks, Claude-Opus-4.5 achieves the strongest overall performance. Under OpenHands, it attains a VS of 58.9 on desktop webpages, a VS/FS of 46.5/66.7 on interactive frontend tasks, and a VS/FS of 38.4/57.6 on full-stack tasks. In contrast, Gemini-3-Pro-Preview and Gemini-3-Flash-Preview perform well on static pages but struggle with long-horizon planning and multi-page integration. Seed-1.8-VL fails entirely on full-stack tasks (VS = 0, FS = 0), while Qwen models largely cannot complete multimodal coding tasks, underscoring substantial disparities in their ability to handle

*Table 3.* End-to-end performance of multimodal coding agents on Vision2Web across three task levels, reporting device-wise static scores, averaged functional scores (FS) and visual scores (VS) for interactive and full-stack tasks, with Deployment Success Rate (DSR) provided for reference rather than an official metric. Unless otherwise noted, all metrics are reported on a 0–100 scale.

| Coding Agent | Static Webpage | | | | | Interactive Frontend | | | | Full-Stack Website | | | |
|---|---|---|---|---|---|---|---|---|---|---|---|---|---|
| | Desktop | Tablet | Mobile | Avg | DSR | VS | FS | Avg | DSR | VS | FS | Avg | DSR |
| **Claude Code** | | | | | | | | | | | | | |
| Claude-Opus-4.5 | 54.2 | 50.4 | 46.8 | 50.5 | 94% | 46.1 | 63.6 | 54.9 | 98.5% | 34.3 | 53.1 | 43.7 | 92.6% |
| Claude-Sonnet-4.5 | 44.7 | 38.6 | 36.7 | 40.0 | 89% | 24.9 | 48.7 | 36.8 | 93.9% | 14.5 | 26.8 | 20.7 | 66.7% |
| GPT-5 | 45.3 | 44.2 | 39.9 | 43.1 | 97% | 23.3 | 64.7 | 44.0 | 98.5% | 9.6 | 23.4 | 16.5 | 85.2% |
| Gemini-3-Pro-Preview | 59.2 | 51.5 | 46.2 | 52.3 | 92% | 13.3 | 14.6 | 14.0 | 65.2% | 5.7 | 12.9 | 9.3 | 63.0% |
| Gemini-3-Flash-Preview | 48.5 | 41.7 | 39.7 | 43.3 | 92% | 13.1 | 27.9 | 20.5 | 78.8% | 2.3 | 4.6 | 3.5 | 63.0% |
| Seed-1.8-VL | 20.1 | 18.6 | 15.3 | 18.0 | 43% | 7.0 | 13.3 | 10.2 | 36.4% | 0.0 | 0.0 | 0.0 | 0.0% |
| Qwen3-VL-32b-Instruct | 0.9 | 1.0 | 0.8 | 0.9 | 3% | 0.0 | 0.0 | 0.0 | 0.0% | 0.0 | 0.0 | 0.0 | 0.0% |
| Qwen3-VL-8b-Instruct | 12.4 | 11.9 | 11.3 | 11.9 | 46% | 0.0 | 0.0 | 0.0 | 0.0% | 0.0 | 0.0 | 0.0 | 0.0% |
| **OpenHands** | | | | | | | | | | | | | |
| Claude-Opus-4.5 | 58.9 | 53.7 | 47.7 | 53.4 | 98% | **46.5** | **66.7** | **56.6** | 98.5% | **38.4** | **57.6** | **48.0** | 96.3% |
| Claude-Sonnet-4.5 | 51.9 | 44.9 | 44.4 | 47.1 | 100% | 32.4 | 59.0 | 45.7 | 97.0% | 15.7 | 23.9 | 20.4 | 66.7% |
| GPT-5 | 49.0 | 44.6 | 40.5 | 44.7 | 100% | 23.9 | 61.4 | 43.2 | 100% | 18.3 | 49.7 | 34.1 | 100% |
| Gemini-3-Pro-Preview | **63.3** | **55.8** | **48.3** | **55.8** | 95% | 29.7 | 40.7 | 35.2 | 93.9% | 11.7 | 22.6 | 17.2 | 77.8% |
| Gemini-3-Flash-Preview | 53.0 | 46.2 | 44.3 | 47.8 | 88% | 25.9 | 38.4 | 32.2 | 93.9% | 7.7 | 17.2 | 12.5 | 66.7% |
| Seed-1.8-VL | 1.1 | 1.4 | 1.4 | 1.3 | 72% | 7.5 | 33.9 | 20.2 | 56.1% | 0.0 | 0.0 | 0.0 | 14.8% |
| Qwen3-VL-32b-Instruct | 0.0 | 0.0 | 0.0 | 0.0 | 4% | 0.0 | 0.0 | 0.0 | 0.0% | 0.0 | 0.0 | 0.0 | 0.0% |
| Qwen3-VL-8b-Instruct | 0.2 | 0.0 | 0.1 | 0.1 | 51% | 0.0 | 0.0 | 0.0 | 0.0% | 0.0 | 0.0 | 0.0 | 0.0% |

complex, multi-stage website development.

**Finding 4:** *Agent performance varies across frameworks.* Framework choice also influences agent performance. Across most models, excluding Claude, performance under the OpenHands framework tends to be higher than under Claude Code, both reflecting framework design differences and indicating that further research on joint optimization of models and agent frameworks could be beneficial for improving overall system performance.

*Table 4.* Performance (Visual Score / Functional Score) of selected coding agents across different website categories under the OpenHands framework in Vision2Web. **Opus-4.5** and **Sonnet-4.5** refer to Claude-Opus-4.5 and Claude-Sonnet-4.5 respectively.

| Website Category | Opus-4.5 | Sonnet-4.5 | GPT-5 |
|---|---|---|---|
| Content | 37.1 / 61.2 | 9.3 / 16.1 | 20.7 / 53.5 |
| Transaction | 43.2 / 64.9 | 10.8 / 14.3 | 13.4 / 50.6 |
| SaaS Platform | 22.9 / 39.9 | 21.7 / 42.8 | 16.7 / 40.5 |
| Public Service | 56.9 / 60.0 | 41.2 / 52.0 | 27.4 / 56.0 |

**Finding 5:** *Full-stack coding performance varies systematically across website categories.*
As shown in Table 4 and further elucidated by qualitative case studies, coding agents exhibit consistent performance differences across website categories. Public Service websites, characterized by simple structures and limited inter-

actions, achieve the strongest visual and functional performance. Content and Transaction websites show moderate performance, reflecting increased presentation or workflow complexity. In contrast, SaaS platforms, which involve multi-page navigation and complex interaction patterns, consistently yield the weakest results, a trend qualitatively associated with higher structural and interaction complexity.

*Table 5.* Performance of selected coding agents under the Open-Hands framework on Vision2Web, reporting category-wise functional scores (average test-case pass rates).

| Test Case Category | Opus-4.5 | Sonnet-4.5 | GPT-5 |
|---|---|---|---|
| Navigation & Routing | 66.3 | 25.9 | 53.9 |
| State Management | 43.2 | 16.1 | 41.5 |
| Form Interaction | 49.2 | 23.7 | 56.8 |
| UI Feedback | 56.4 | 23.1 | 30.8 |
| Authentication & Authorization | 61.8 | 25.5 | 60.0 |
| CRUD Operations | 43.6 | 20.5 | 35.9 |
| Search / Filter / Sorting | 55.6 | 16.7 | 50.0 |
| File & Media Operations | 33.3 | 0.0 | 16.7 |

**Finding 6:** *At the level of individual functional categories, agents exhibit systematic weaknesses in complex, state-dependent operations.*
When examined at the level of individual test-case categories, Navigation & Routing and Authentication & Au-

thorization are the most reliable capabilities across models, with Claude-Opus-4.5 and GPT-5 achieving consistently high pass scores. In contrast, performance drops markedly on State Management, CRUD Operations, and File & Media Operations. These tasks demand persistent state tracking, correct data flow across components, or coordination between frontend logic and system-level resources, which remain challenging even for the strongest models.

### 4.3. Analysis of Failure Modes

We analyze representative failure cases on Vision2Web across hierarchical task levels, revealing distinct capability gaps as development complexity increases.

**Fine-Grained Visual Alignment Failures.** At the lowest level, agents often fail to reproduce fine-grained visual details, including misaligned layouts, incorrect sizes, and color mismatches, especially for regularly arranged components. Asset handling is particularly fragile: agents over-rely on file names and lack robust multimodal grounding, causing visible inconsistencies when assets are unnamed or ambiguously referenced, even on static webpages.

**Cross-Module Visual Understanding Failures.** Errors intensify when tasks involve multiple modules or pages. While homepages are typically reproduced reasonably well, visual fidelity degrades on subsequent pages, with missing or misaligned components, malfunctioning interactive elements, and broken navigation links. These failures reflect difficulty in maintaining coherent visual and functional reasoning across views.

**System-Level Integration and State-Consistency Failures.** At the full-stack level, failures show a capability-dependent pattern. Weaker agents often fail before integration can be meaningfully tested, due to incomplete modules, missing pages, deployment failures, or runtime crashes. Stronger agents can generate largely complete projects, and their remaining errors shift toward integration and state consistency, including incomplete database persistence, incorrect API routing and inconsistent frontend–backend data flow. Across models, fine-grained requirement following remains a shared weakness: agents often implement the common path while omitting boundary cases, exceptional states, or specific constraints. These results suggest that current coding agents struggle not only with code generation, but also with maintaining specification fidelity and system consistency over long implementation trajectories.

### 4.4. Validation of the Agent Verifier

We assess the reliability of both verifiers by measuring agreement with human annotations for the GUI agent and rank consistency with human preferences for the VLM-based judge.

**GUI Agent Verifier Validation.** We randomly sample approximately 100 test workflows from 64 tasks. For each workflow, all constituent test nodes are independently examined by human annotators to verify whether they satisfy the intended test requirements. At the node level, 218 of 250 nodes (87.2%) are correctly judged by the verifier relative to human annotations, indicating high fine-grained execution accuracy. Residual inaccuracies are largely attributable to model-intrinsic reasoning hallucinations, which are expected to diminish as the proficiency of the GUI Agent continues to improve.

**VLM-Based Judge Validation.** We evaluate the consistency between the VLM-Based Judge and human judgments using the Spearman rank correlation coefficient ($\rho$), a standard metric for evaluating preference alignment and ranking consistency in subjective judgment tasks (Gu et al., 2026), with $\rho > 0.5$ indicating substantial rank consistency. Across 100 randomly sampled prototypes, the VLM-Based Judge achieves an average Spearman correlation of 0.66, with a median of 0.80, while human inter-annotator agreement on the same set yields a Spearman correlation of 0.78. Given the intrinsic subjectivity of visual preference judgments and the non-trivial disagreement among human annotators, the observed correlation represents a strong and practical level of alignment in most instances, while leaving room for improvement in challenging cases.

*Table 6.* Sensitivity of functional verification to GUI verifier backbones on Level-3 website tasks. FS denotes Functional Score.

| Tested Model | GLM-4.6V FS | GPT-5-Mini FS |
|---|---|---|
| Claude-Sonnet-4.6 | 48.6 | 46.6 |
| GPT-5.4 | 28.6 | 29.3 |
| Gemini-3.1-Pro-Preview | 17.0 | 19.1 |
| Overall | 31.36 | 31.67 |

**Sensitivity Analysis.** We further analyze whether the evaluation results are sensitive to the choice of verifier backbones. For visual evaluation, we re-score 567 prototypes from 153 projects using four VLM judges: Gemini-3.1-Pro-Preview (Google, 2026b), Doubao-Seed-2.0-Pro (ByteDance Seed Team, 2026), GLM-4.6V, and GPT-5.4 (OpenAI, 2026). Although the absolute score calibration varies across judges, all four judges produce the same system-level ranking: Claude-Sonnet-4.6 (Anthropic, 2026b) > GPT-5.4 > Kimi-K2.5 (Kimi Team et al., 2026) > Gemini-3.1-Pro-Preview, suggesting that the visual evaluation is robust at the ranking level.

For functional evaluation, we compare two GUI verifier backbones, GLM-4.6V and GPT-5-Mini-2025-08-07 (OpenAI, 2025b), on Level-3 website tasks. As shown in Table 6, the two verifiers yield nearly identical overall functional scores, with an absolute difference of 0.31 and a relative difference of 1.0%. They also preserve the same model

ranking across evaluated systems. These results indicate that the main conclusions of Vision2Web are not driven by a single verifier backbone.

To further maintain evaluator reliability over time, we plan to update both the VLM judge and GUI agent on a quarterly basis using the latest backbone models.

## 5. Related Work

### 5.1. UI2Code

UI-to-code generation has advanced through benchmarks and datasets that map visual layouts to executable code. Early works, such as Design2Code (Si et al., 2025), introduced automated metrics like Block-Match and CLIP similarity. Subsequent efforts, including Web2Code and Flame-React, expanded datasets from synthetic resources like WebSight (Laurençon et al., 2024) to real-world collections such as WebCode2M (Gui et al., 2025). More recent work further extends this line beyond one-shot screenshot-to-code generation: UI2Code$^N$ (Yang et al., 2025) formulates UI-to-code as an interactive visual optimization process with rendered feedback, while WebVR (Dai et al., 2026) studies webpage recreation from demonstration videos using human-aligned visual rubrics. Despite these advances, most benchmarks still focus primarily on visual recreation, static layouts, or short-horizon refinement, limiting systematic evaluation for complex, interactive, real-world websites.

### 5.2. Autonomous Coding Agents

Autonomous coding agents evolved from single-shot code generation to multi-step, interactive systems. Early work enhanced agents with planning, reasoning, and iterative refinement (Self-Planning (Jiang et al., 2024), CodeChain (Le et al., 2023), CodeAct (Wang et al., 2024)), while later agents integrated tool use, retrieval, and execution feedback for robustness (ToolCoder (Zhang et al., 2023), CodeAgent (Zhang et al., 2024)). Modern practical agents like Copilot (GitHub, 2023), Cursor (Cursor, 2026), and Claude Code (Anthropic, 2026a) support multi-file refactoring and end-to-end software development.

### 5.3. Evaluation of Coding Agents

Early evaluations of code generation primarily focused on file- or function-level tasks, using benchmarks such as HumanEval (Chen et al., 2021) and MBPP (Austin et al., 2021), and later programming contest datasets including APPS and LiveCodeBench (Jain et al., 2025), where models were assessed mainly on functional correctness in isolated contexts. More recently, real-world software development benchmarks such as SWE-Bench and its variants (Jimenez et al., 2024; Yang et al., 2024) evaluate agents' abilities to navigate large codebases, interact with tools, and iteratively resolve complex issues. Complementing these, emerging evaluations including WebGen Bench (Lu et al., 2025) and VIBE Bench (MiniMax, 2025) extend assessment to end-to-end, from-scratch project development. However, existing benchmarks remain limited by the lack of visual-centric coding tasks for evaluating cross-modal reasoning, insufficiently structured hierarchical task inputs for comprehensive measurement, and coarse end-to-end evaluation criteria that hinder reliable and reproducible assessment.

## 6. Conclusion

We present Vision2Web, a comprehensive benchmark for evaluating multimodal coding agents in visual-centric website development. By organizing tasks into three hierarchical levels, Vision2Web enables systematic assessment under increasing task complexity. The benchmark introduces a workflow-based agent verification paradigm, combining a GUI agent verifier with a VLM-based judge, allowing reproducible, holistic measurement of functional correctness and visual fidelity. Large-scale experiments show that strong performance on isolated tasks does not reliably transfer to end-to-end system construction, revealing systematic deficiencies in handling structural complexity, cross-page coordination, and persistent state reasoning. These findings call for a shift toward hierarchical, progressively challenging task designs and principled, reproducible autonomous evaluation paradigms as the foundation for rigorously understanding and assessing the capabilities of coding agents.

## Limitations

Several limitations should be considered when interpreting Vision2Web. Its evaluation relies on automated GUI-agent execution and VLM-based visual judgment, which may still introduce residual errors despite our human validation and verifier-sensitivity analyses. In addition, Vision2Web focuses on end-to-end website development rather than the full spectrum of software engineering tasks. Finally, realistic multimodal evaluation is more costly than code-only metrics, since it requires verifying deployed behavior, visual fidelity, and user-facing functionality. This reflects a broader trend in agent evaluation: as benchmarks move toward end-to-end long-horizon system construction, verification is becoming richer, more realistic, and closer to deployed behavior.

## Impact Statement

Our benchmark is constructed entirely from publicly accessible websites and other openly available resources. All data is used solely for academic research purposes, and no private, sensitive, or personal information is included. There

are no associated negative ethical or legal impacts, and the benchmark is intended to provide a reproducible and controlled framework for evaluating and advancing the field of Machine Learning.

## Acknowledgments

This work was supported by the Natural Science Foundation of China (62425601), Fundamental and Interdisciplinary Disciplines Breakthrough Plan of the Ministry of Education of China (No. JYB2025XDXM101), the new cornerstone Science Foundation through the XPLORER PRIZE and a research fund from Daimler Greater China Ltd., Tsinghua University Joint Institute for Sustainable Mobility, the National Natural Science Foundation of China (62506195), China Postdoctoral Science Foundation (2025M771572) and China Postdoctoral Program for Innovative Talents (BX20250381).

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

# A. Appendix.

## A.1. Benchmark Details and Statistics

### A.1.1. WEBSITE CATEGORY DISTRIBUTION

To characterize website diversity in Vision2Web, we construct a professional website taxonomy through interviews with experienced frontend engineers and synthesis by PhD students in computer science. The taxonomy groups websites into four broad classes—Content, Transaction, SaaS Platforms, and Public Services—with subcategories, representative examples, and associated software engineering capabilities summarized in Table 7.

*Table 7.* Website categories in Vision2Web, subcategory definitions, and representative examples.

| Macro Category | Subcategory | Description | Examples |
|---|---|---|---|
| Content | News | Platforms for publishing timely news content | CNN, BBC, NYTimes |
| | Community | Platforms for content and social interaction | Reddit, Zhihu, StackOverflow |
| | Multimedia | Platforms for consuming rich media content | YouTube, Spotify, Vimeo |
| | Knowledge | Platforms for knowledge delivery | Coursera, Khan Academy, edX |
| | Publishing | Platforms for content creation and presentation | Medium, WordPress, Substack |
| Transaction | Retail | Platforms for consumer product transactions | Amazon, Taobao, JD |
| | Vertical Markets | Platforms for domain-specific marketplaces | Airbnb, Booking.com, Xianyu |
| | Services | Platforms for online service interactions | PingAn Insurance |
| | Procurement | Platforms for enterprise-level purchasing | Alibaba B2B, Global Sources |
| SaaS Platforms | CRM | Platforms for customer data management | Salesforce, HubSpot |
| | HCM | Platforms for managing human capital | Workday, SAP SuccessFactors |
| | ERP | Platforms for enterprise resource | Jira, Confluence |
| | TMS | Platforms for task management | Trello, ClickUp, Notion |
| | AI Platform | Platforms providing access to AI service | OpenAI Playground |
| Public Services | Government Portal | Official platforms for government services | Gov.cn, IRS, GOV.UK |
| | Public Utility Websites | Platforms supporting essential public services | Education bureaus |

Building upon this classification, we further analyze how tasks in Vision2Web are distributed across different website categories and task levels. Table 8 summarizes the number of tasks in each category under the three hierarchical development levels, providing an overview of the benchmark composition and ensuring balanced coverage across both website types and software engineering complexity.

### A.1.2. TASK-LEVEL STATISTICS

The following presents the feature distribution of tasks across different levels. For static webpage tasks, Vision2Web exhibits higher difficulty compared to traditional Design2Code benchmarks, covering a wider range of prototype sizes.

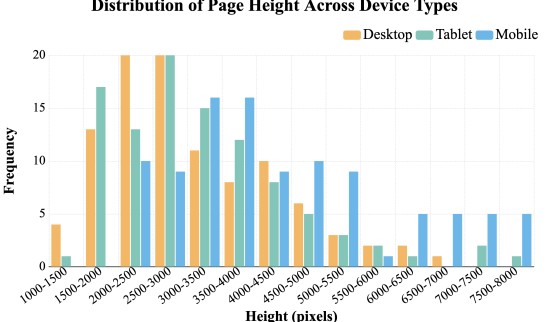

*Figure 5.* Distribution of prototype image sizes across different device types.

| Dataset | Design2Code-Hard | Vision2Web |
|---|---|---|
| Avg Tag Count | $251 \pm 232$ | $1385 \pm 985$ |
| Avg DOM Depth | $10 \pm 4$ | $22 \pm 7$ |
| Avg Unique Tags | $22 \pm 5$ | $40 \pm 14$ |

*Figure 6.* Comparison of task complexity metrics between Design2Code-Hard and Webpage tasks of Vision2Web.

*Table 8.* Distribution of tasks across website categories and development levels in Vision2Web.

| Macro Category | Subcategory | Static Webpage | Interactive Frontend | Full-Stack Website |
|---|---|---|---|---|
| Content | News | 8 | 7 | 2 |
| | Community | 8 | 6 | 2 |
| | Multimedia | 6 | 5 | 2 |
| | Knowledge | 10 | 7 | 2 |
| | Publishing | 8 | 5 | 2 |
| Transaction | Retail | 10 | 5 | 2 |
| | Vertical Markets | 8 | 6 | 2 |
| | Services | 6 | 3 | 2 |
| | Procurement | 4 | 4 | 2 |
| SaaS Platforms | CRM | 3 | 3 | 1 |
| | HCM | 3 | 2 | 1 |
| | ERP | 5 | 1 | 1 |
| | TMS | 3 | 3 | 1 |
| | AI Platform | 3 | 2 | 1 |
| Public Services | Government Portal | 9 | 4 | 2 |
| | Public Utility Websites | 6 | 3 | 2 |

### A.1.3. REPRESENTATIVE TASK EXAMPLES

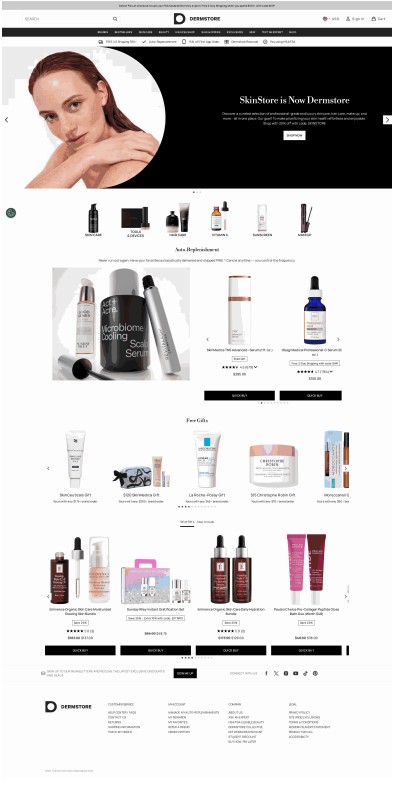
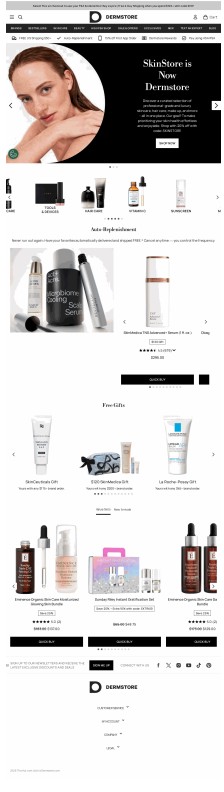
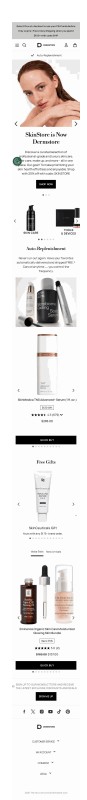

*(a)* Desktop        *(b)* Tablet        *(c)* Mobile

*Figure 7.* Cross-device responsive static webpage task example.

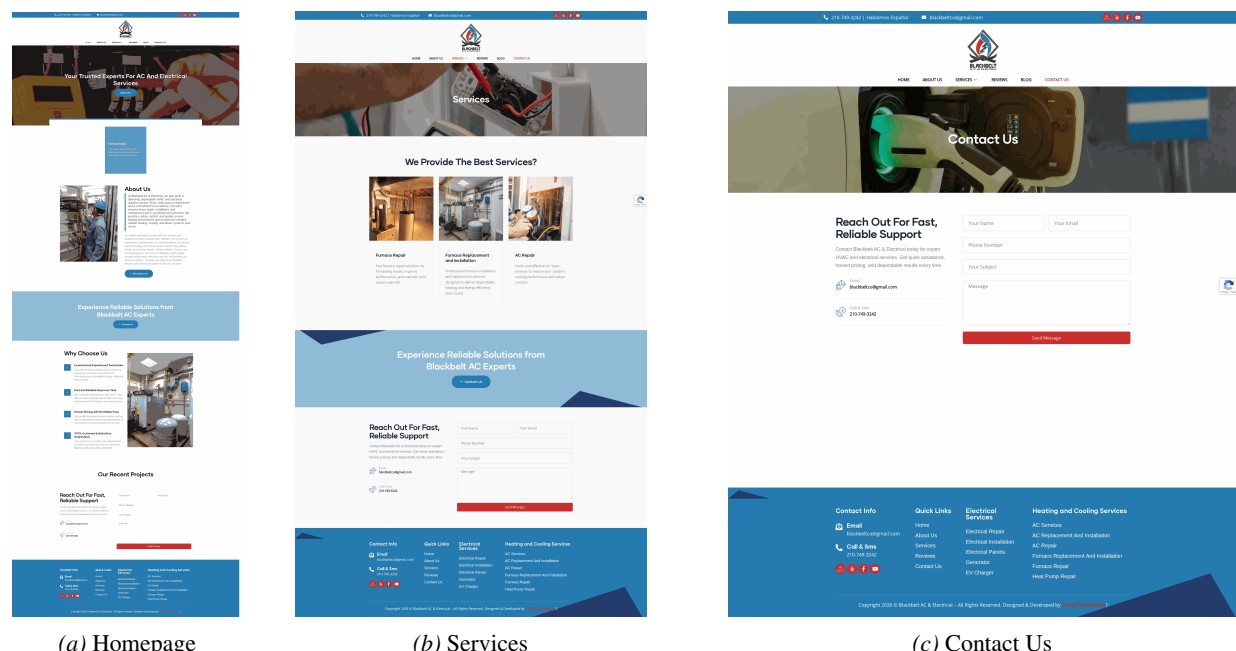

*(a)* Homepage  *(b)* Services  *(c)* Contact Us

*Figure 8.* Interactive frontend task example.

---

**Simplified Textual Description Example for Interactive Frontend Tasks**

I want to build a Blackbelt AC & Electrical website, a multi-page service company site with a global navigation bar (Home, About Us, Services with dropdown, Reviews, Blog, Contact Us) and footer quick links. The homepage features a hero section with call-to-action buttons and service showcase cards, while dedicated pages cover company information, categorized AC and electrical services, detailed service pages with descriptions and contact forms, blog listings and posts, and a contact page with forms and social links. Users can navigate across pages via the menu or footer, explore services and blog content, and submit inquiries.

---

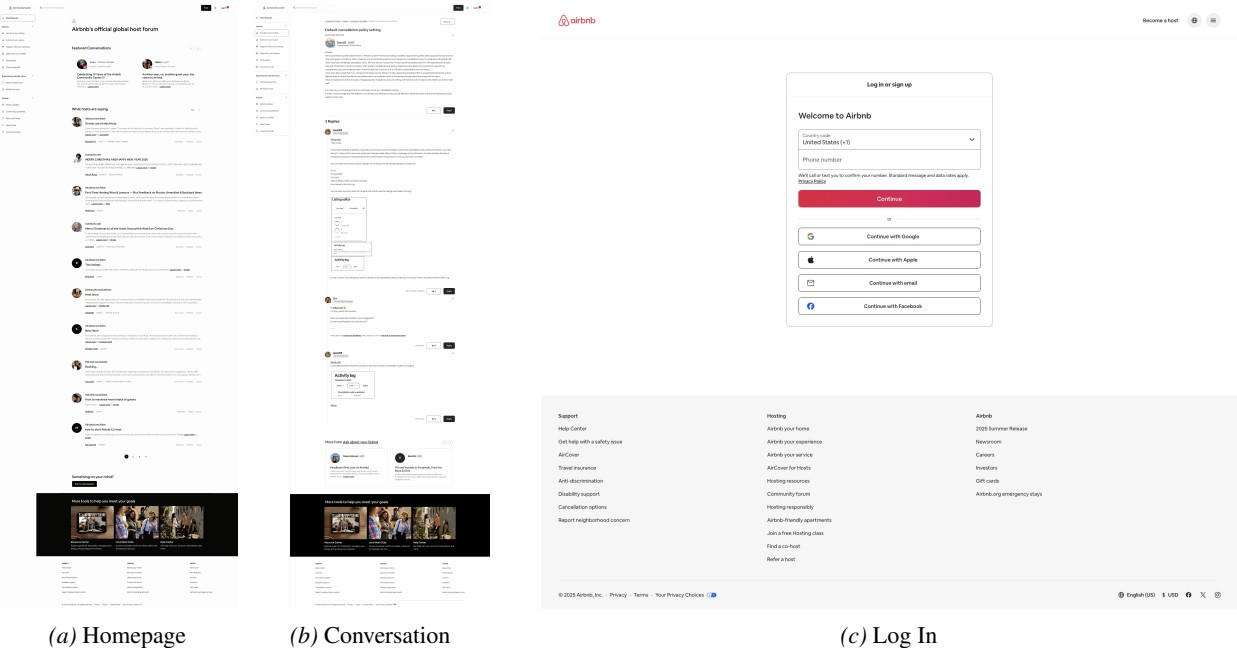

*(a)* Homepage  *(b)* Conversation  *(c)* Log In

*Figure 9.* Full-stack website task example.

**Simplified Requirement Document Example for Full-Stack Website Tasks**

## 1. Product Background

The Airbnb Community Center is a forum for hosts and community members to exchange experiences and seek advice on topics like property management, booking support, and hosting services.

## 2. Business Workflows

### 2.1 Page Navigation Flow

Pages: Home, Login/Signup, Topic, Post Detail, Post Creation, User Profile. Navigation via top bar, sidebar categories, featured posts, user avatars, and action buttons. Supports login-dependent and unauthenticated flows.

### 2.2 Core Workflows

- **User Authentication:** Login via phone number → Validate → Return home
- **Browse & Discovery:** Home/Topic pages → View featured posts → Filter by tags
- **Posting & Interaction:** Create post → View details → Reply & like interactions
- **User Profile Access:** Visit own or others' profiles → View stats and activity records

## 3. Requirements Specification

### 3.1 User Authentication

- Phone number login with validation
- Login state persistence and avatar display

### 3.2 Home Portal

- Top nav with logo, search, Post button, language toggle
- Featured conversations & popular post list
- Sidebar category navigation
- Post creation button with permission control

### 3.3 Topic Category

- Breadcrumb, title, description
- Featured posts and post list with tag filtering
- Pagination support
- Start a conversation button with permission control

### 3.4 Post Detail

- Breadcrumb, title, timestamp
- Author info + post content
- Reply & like interactions (login-dependent)
- Reply list display
- Related post recommendations

### 3.5 Post Creation

- Page access restricted to logged-in users
- Title, content, topic/sub-topic selection (required)
- Optional tag selection
- Submit & cancel actions with validation

### 3.6 User Profile

- Display avatar, name, level, location
- Show post/reply/like counts
- Activity records list (posts & replies)

## A.2. Workflow-Based Agent Verification Implementation Details

### A.2.1. ILLUSTRATIVE WORKFLOW EXAMPLE

We present a representative workflow instance from our dataset to illustrate how end-to-end GUI testing tasks are structured and executed. Each workflow consists of a sequence of objectives, guided actions, and validation criteria, executed sequentially within a single shared browser session.

---

**Simplified Workflow Example**

**Summary:** Post creation form validation testing with positive and negative scenarios for title, topic selection, sub-topic selection, and content fields

**Resolution:** $1920 \times 1080$

- **Objective:** Navigate to the post creation page as a logged-in user.
  **Actions:**
  1. Click the Log in button in the navigation bar
  2. Type "12345678" into the Phone number field
  3. Click the Continue button
  4. Click the Post button in the navigation bar

  **Validations:** None

- **Prototype:** post

- **Objective:** Verify that the post creation form accepts valid input for title, topic selection, and sub-topic selection fields.
  **Actions:**
  1. Type "Test topic" into the Enter the topic here field
  2. Select "Homes" from the topic dropdown
  3. Click the sub-topic card labeled "Advice on your space"

  **Validations:**
  – The title field displays "Test topic"
  – The topic dropdown displays "Homes"
  – The sub-topic card "Advice on your space" is visually selected

- **Objective:** Verify that post submission fails when the content field is empty, demonstrating a negative test scenario for required field validation.
  **Actions:**
  1. Click the Submit button

  **Validations:**
  – Post submission fails and does not proceed
  – An error message is displayed indicating that the content field is required

- **Objective:** Verify that post creation succeeds when all required fields are filled with valid data, and the user is navigated to the newly created post detail page with matching content.
  **Actions:**
  1. Type "It's a test content" into the content field
  2. Click the Submit button

  **Validations:**
  – Post submission succeeds

---

> - The page navigates to the newly created post detail page
> - The post title displays "Test topic"
> - The post content displays "It's a test content"

### A.2.2. AGENT CONFIGURATION

The GUI agent and the VLM judge are instantiated with GLM-4.6V (GLM-V Team et al., 2025) and Gemini-3-pro-preview (Google, 2026a), respectively. We present the prompts used for agent verification below, showing their overall structure with certain details simplified.

---

**Structure of GUI Agent Verifier Prompt**

You are a **GUI Testing Agent**.
Your primary task is to **execute software test cases on a Web application** by interacting with the graphical user interface and determining whether the test case **passes or fails** based on defined validation criteria.
The current time is {time}.

**Context:**
{context}

## Test Case
**Objective**
{objective}

**Actions**
{actions}

**Validations**
{validations}

## Test Platform
Web

## Action Space
The agent operates in a predefined GUI action space (Click, Type, Scroll, Wait, GoBack, Refresh, Key, Answer), following standard web interaction semantics.

## History
You have **already performed the following actions** (format: Thought, Action):
{history}

## Output Format
Your reply should strictly follow the format:
- Thought: Your brief thoughts
- Action: One Action format you choose

## Current Observation
Full-page screenshot: {fullpage_screenshot}
Current viewport screenshot: {viewport_screenshot}

---

---

### Structure of VLM Judge Prompt

You are a **senior QA automation engineer**. Your task is to **compare a prototype image with an actual page screenshot** and score the appearance of the page.

**Important:** Carefully observe **all differences** between the prototype and the actual page. Do not overlook any discrepancies, however small. Score strictly according to the rules below.

#### Input
1. **Prototype Image**: {prototype_page}
2. **Actual Page Image**: {actual_page}

*Note:* You need to automatically segment the page into meaningful UI components based on visual and functional layout.

#### Component Segmentation Rules
- Divide the page into **logical functional blocks**, not too granular or too coarse.
- For each block, treat it as a **single component** for scoring purposes.

#### Scoring Rules
Each component or block receives an independent score:

| Score | Description |
|---|---|
| 1.0 | **Perfect Match:** Component position exactly matches the prototype. Layout, spacing, alignment, and sizing are identical. Text, fonts, colors, icons, and images are fully accurate. No visually discernible differences. |
| 0.75 | **Minor Imperfections:** Mostly accurate positioning with very slight misalignment ($< 2px$). Layout and spacing largely consistent. Only minor typos or formatting differences. Multimedia shows slight scaling or color variation. |
| 0.5 | **Partial Match:** Roughly correct position but noticeable misalignment or spacing issues. Layout partially consistent. Multiple text discrepancies. Multimedia partially incorrect or inconsistent. |
| 0.25 | **Poor Match:** Component recognizable but significantly misaligned. Layout mostly inconsistent. Text differs significantly. Multimedia missing or incorrect. |
| 0.0 | **No Match:** Component missing or completely misplaced. Layout unrelated to prototype. Text and multimedia absent or entirely incorrect. |

#### Output Requirements
Output **only** a JSON object in the following structure:

```
[
  {
    "name": "<component name>",
    "score": <0 | 0.25 | 0.5 | 0.75 | 1>,
    "reason": "<brief explanation of why this score was given>"
  }
]
```

---

## A.3. Experimental Setup and Prompts

### A.3.1. AGENT PROMPT TEMPLATES

Our evaluation prompt is carefully constructed to (1) explicitly enumerate all available input materials, including UI prototypes and textual requirements; (2) specify the expected level of task completion corresponding to the task hierarchy; and (3) discourage premature termination or unnecessary over-engineering. Prompt templates are shown below:

---

**Structure of Full-Stack Task Prompt**

---

You are a **senior full-stack engineer with expertise in web development**.

Your task is to **implement and deploy a production-ready web application** strictly following the provided materials in `/workspace`.

**I. Input Materials**

1. **Product Requirement Document** (`/workspace/prd.md`): Contains overview, business logic, and detailed requirements.
2. **Prototype Images** (provided above, not from files): Define UI layout, style, interactions, and visual fidelity.
3. **Resource Files** (`/workspace/resources/**/*`): Assets including images, videos, audio, icons, etc.

**II. Mandatory Full-Stack Workflow**

1. **Planning & Design**
   - Define database schema and seed data.
   - Specify front-end/back-end architecture and tech stack.
   - Design project directory structure.

2. **Seed Data Generation**
   - Generate complete realistic seed data reflecting prototypes.
   - Populate all UI elements and states (loading, empty, error).

3. **Full-Stack Implementation**
   - **Back-End:** APIs, validation, authentication, database initialization.
   - **Front-End:** Replicate visuals, implement all interactions, consume live APIs.
   - **Integration:** Ensure front-end and back-end work seamlessly.

4. **Deployment & Verification**
   - Build application and verify accessibility at `http://localhost:3000`.
   - Generate `/workspace/start.sh` for fully reproducible deployment.
   - Verify startup script in a clean environment.

5. **Documentation**
   - Design document: architecture, data model, technology stack.
   - README: overview, stack, directory, deployment instructions.

**III. Required Deliverables**
- Complete source code
- Design document
- Seed data
- Deployment script (`start.sh`)
- README

**IV. Hard Constraints**
- Implement all PRD features without skipping or inventing functionality.
- All visuals and interactions must match prototypes.
- System must be fully reproducible via `bash /workspace/start.sh`.
- Do not terminate the work process until all steps and startup verification are complete.

---

A.3.2. TOOLS AND RESOURCE CONSTRAINTS

The agent evaluation environment is configured with only the necessary foundational tools required for standard software development. This includes a terminal tool for command execution, basic file read/write operations. No Model Context Protocol (MCP) or additional orchestration/configuration layers are installed, ensuring that the agent relies solely on its own reasoning and coding capabilities to complete the assigned tasks.

Our inference and evaluation container includes the necessary runtimes, system libraries, and development tools, while

preserving file system access for the agent's workspace. Key configuration details are summarized in the Dockerfile listed below, which defines the base operating system, programming environments, database clients, and essential system utilities required for full-stack website development tasks.

```
1   FROM ubuntu:22.04
2
3   # Locale
4   RUN locale-gen en_US.UTF-8
5   ENV LANG=en_US.UTF-8 LC_ALL=en_US.UTF-8
6
7   # Essential system tools & database clients
8   RUN apt-get update && apt-get install -y \
9       curl wget git vim unzip sudo build-essential \
10      gcc g++ make cmake net-tools iputils-ping \
11      postgresql-client default-mysql-client redis-tools sqlite3 \
12      && rm -rf /var/lib/apt/lists/*
13
14  # Node.js 20.x
15  RUN curl -fsSL https://deb.nodesource.com/setup_20.x | bash - \
16      && apt-get install -y nodejs
17
18  # Python 3.12
19  RUN add-apt-repository ppa:deadsnakes/ppa -y \
20      && apt-get update && apt-get install -y \
21          python3.12 python3.12-venv python3-pip \
22      && update-alternatives --install /usr/bin/python3 python3 /usr/bin/python3.12 1
23
24  # Create agent user & workspace
25  RUN useradd -m -s /bin/bash agent \
26      && mkdir -p /workspace \
27      && chown -R agent:agent /workspace
28  USER agent
29  WORKDIR /workspace
30
31  # Python packages for agent evaluation
32  RUN python3 -m pip install --user --upgrade pip setuptools wheel
33  RUN python3 -m pip install --user playwright claude_agent_sdk==0.1.18 openhands
34  RUN python3 -m playwright install chromium
35
36  # Default command
37  CMD ["/bin/bash"]
```

## A.4. Additional Results and Analysis

### A.4.1. ILLUSTRATIVE FAILURE CASES

Below, we present a systematic analysis of representative failure cases across the three task levels in Vision2Web. Rather than isolated errors, these failures arise at successive stages of the website development process, where increasingly demanding requirements on visual perception, cross-module understanding, and long-horizon planning progressively expose fundamental limitations of current multimodal coding agents.

At lower levels, agents struggle to ground fine-grained visual details into precise layout and styling decisions, leading to misalignment and visual inconsistencies. As task scope expands to multiple components and pages, these errors compound due to insufficient cross-page state tracking and weak integration of heterogeneous visual and textual cues. At the system level, the absence of reliable self-verification and planning mechanisms causes accumulated deviations from specifications, ultimately resulting in breakdowns in functional correctness and execution stability.

By tracing failures from fine-grained visual reproduction to cross-page coherence and full-system execution, we reveal how agent performance systematically degrades as task complexity and dependency structure increase.

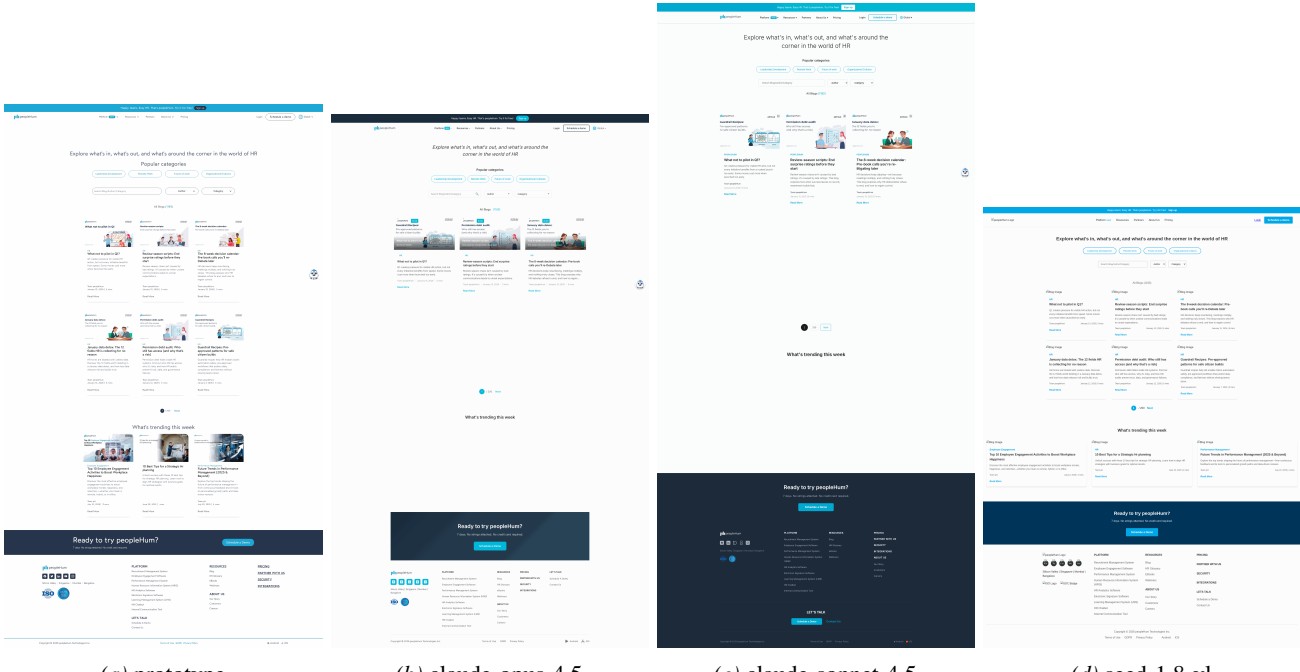

*(a)* prototype          *(b)* claude-opus-4.5          *(c)* claude-sonnet-4.5          *(d)* seed-1.8-vl

*Figure 10.* failure cases of static webpage tasks

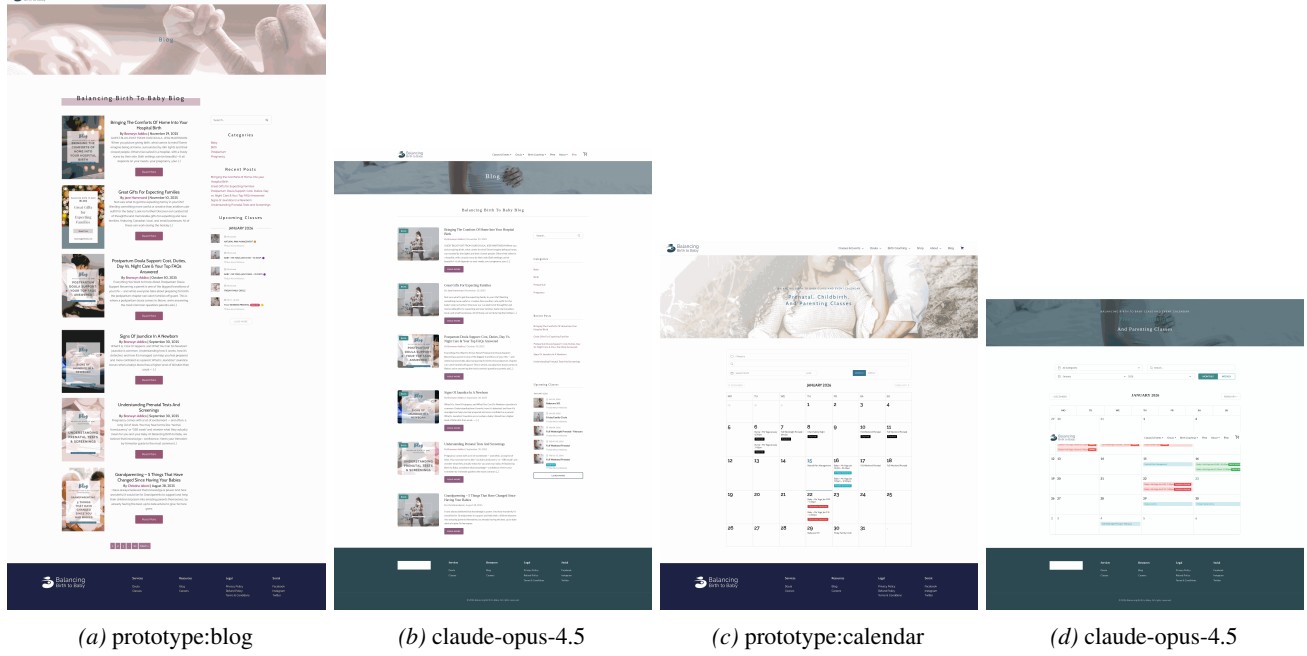

*(a)* prototype:blog          *(b)* claude-opus-4.5          *(c)* prototype:calendar          *(d)* claude-opus-4.5

*Figure 11.* failure cases of interactive frontend tasks

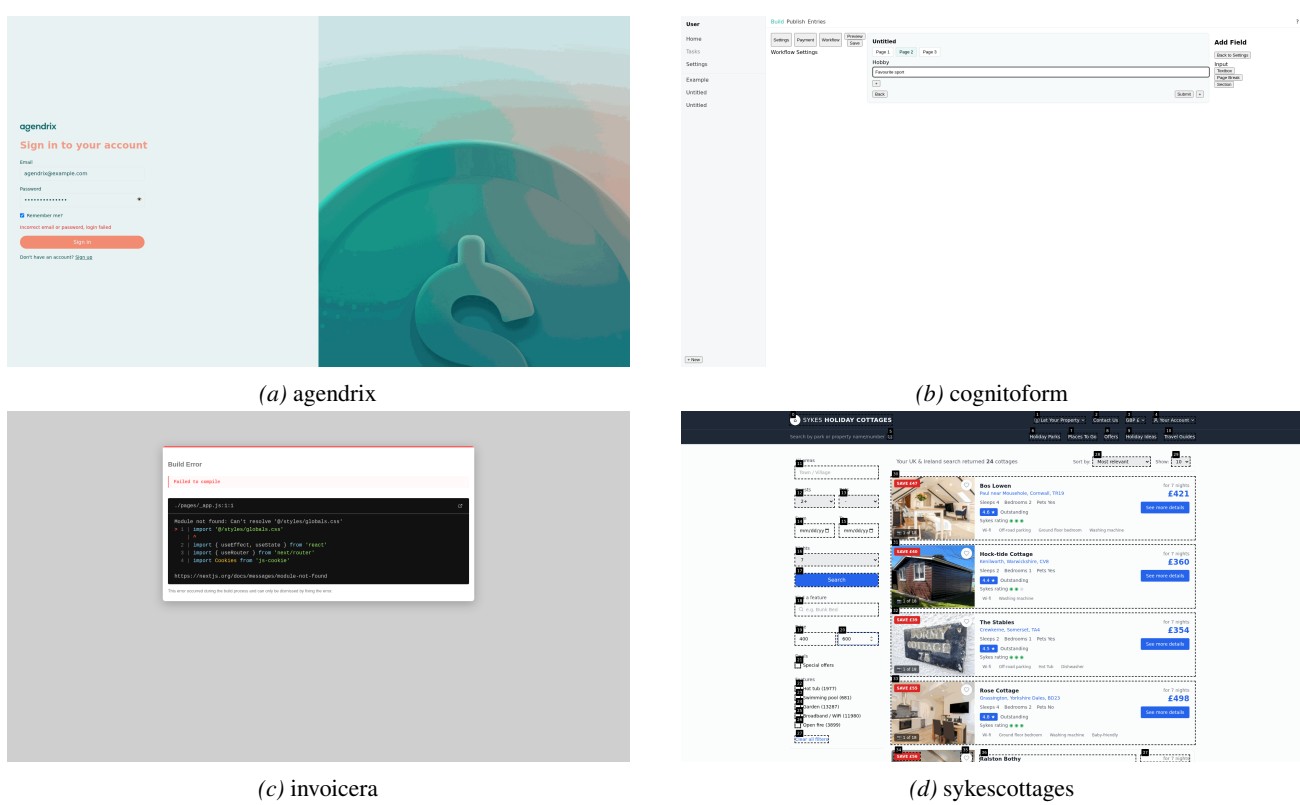

*(a)* agendrix

*(b)* cognitoform

*(c)* invoicera

*(d)* sykescottages

*Figure 12.* failure cases of full-stack website tasks

