# OpenReview forum: "Vision2Web: A Hierarchical Benchmark for Visual Website Development with Agent Verification"
_ICML.cc/2026/Conference — ICML 2026 spotlight_

### Official Review · Reviewer_oBq1 · 2026-02-20

**Soundness:** 3
**Presentation:** 3
**Significance:** 4
**Originality:** 4
**Overall Recommendation:** 5
**Confidence:** 4

**Summary:**

This paper built a test called VisionWebDev that collected real websites, organized development tasks into three difficulty levels (static website, interactive frontend, and Full-Stack Website), and created a rigorous automated system to evaluate what the coding agent produces. The authors wanted to provide a better testing process for AI coding agents since existing ones only focus on simple tests like whether an AI can fix a small bug in an existing code or reproduce a single static webpage.
To evaluate, they used a two-pronged checking system ( a GUI Agent Verifier and a VLM judge) that produces two scores for each. The GUI Agent Verifier verifies that the features of the website actually work, while the VLM judge compares the generated site against the original designs.
When they tested several leading AI models on this benchmark, they found that all of them struggled significantly as tasks got harder, and Claude-Opus-4.5 performed best overall.

**Compliance With Llm Reviewing Policy:**

Affirmed.

**Final Justification:**

The authors have done well with their response, i will keep the score unchanged

**Key Questions For Authors:**

None

**Limitations:**

The authors have not adequately discussed limitations. The paper will greatly benefit from a dedicated limitations section

Claude Code was used for annotation, while Claude-Opus-4.5 and Claude-Sonnet-4.5 are among the models being tested on the very benchmark that Claude Code helped build and annotate. which could be a possible reason why Claude-Opus-4.5 was the best performing model.
Secondly, this experiment must have been expensive to run. Talking about this will help future researchers know the implications of doing the project

**Strengths And Weaknesses:**

Soundness
The paper is technically sound. However, Claude-Opus-4.5 and Claude-Sonnet-4.5 are among the models being tested on the very benchmark that Claude Code helped build and annotate, which could be a reason Claude-Opus-4.5 performed best.

Presentation
The narrative is easy to follow. However, the paper has a few structural issues.
All Figures except Figure 1 are hard to read unless you zoom in, especially 2,3,4 since they show important graphs (the rest can be ignored)
The paper is organized differently from the norm (Introduction -> related work -> Method -> Results -> Discussion -> Conclusion). The related works section comes at the end of the paper, but somehow, prior work was discussed in the introduction, Overview of VisionWebDev, and the related work section. Getting to the end of the paper, there's no need for the related work section anymore.

Significance
This is very Important work! As AI coding agents are being deployed in real software development contexts, having a benchmark that tests end-to-end, full-stack capability rather than isolated skills is a meaningful step forward. The finding that even the best models struggle significantly on full-stack tasks is a useful signal for the research community. The benchmark could become a standard reference point for evaluating future coding agents, but real websites change over time, which means the benchmark could slowly become outdated without anyone noticing.

Originality
The originality lies in the combination of the individual pieces (GUI agents, visual judges, hierarchical tasks), which come together to form something truly meaningful. No weaknesses come to mind

---

> ### Author Rebuttal · Authors · 2026-03-31
>
> We sincerely thank the reviewer for the highly positive assessment of the paper’s significance and originality. We address the main concerns below.
>
> ### 1. Claude Code involvement and potential annotation bias
>
> **Claude Code played a limited assistive role in annotation refinement, rather than determining task specification or evaluation criteria.** Task design, specifications, and verification workflows were primarily defined by human experts based on original websites. Claude Code was limited to structuring and refining human-authored annotations into structured verification workflow formats; it did not define task requirements, set scoring criteria, or determine evaluation process. We will clarify this role more explicitly and concretely in the revision.
>
> ### 2. Limitations section
>
> We agree that making these limitations explicit will better contextualize the benchmark’s intended use, without changing its core contribution. In the revision, we will add this section and explicitly discuss at least three points:
> (1) **evaluator imperfection**, including the current reliance on GUI-agent execution and VLM-based visual judgment;
> (2) **the scope of the benchmark**, which focuses specifically on end-to-end website development rather than the full spectrum of software engineering tasks;
> (3) **evaluation cost trade-offs**, realistic end-to-end multimodal assessment requires richer verification than code-only metrics;
>
> ### 3. Experimental cost and practical implications
>
> **Richer evaluation is necessary for realistic end-to-end agent assessment, and this inevitably increases evaluation cost.** Because VisionWebDev targets long-horizon multimodal development, its evaluation covers interaction, coordination, state, backend functionality, and visual fidelity. This makes it more expensive than code-only metrics, but also yields more valuable evaluation insights for realistic end-to-end assessment.
>
> **In practice, the cost remains feasible.** A full single-model run uses roughly 5–10M tokens for the VLM judge and 50–80M tokens for the GUI verifier, and can generally be completed within one day. We further reduce cost through prompt-cache reuse, tighter GUI-agent context control. We will make these cost implications more explicit in the revision so future researchers can better understand the implications involved.
>
> ### 4. Benchmark maintenance over time
>
> **The benchmark’s core target capabilities remain meaningful even as individual websites evolve over time.** Multimodal grounding, long-horizon planning, cross-module coordination, and full-stack integration remain meaningful evaluation targets even as specific websites evolve over time.
>
> **To keep the benchmark timely, we plan to maintain and periodically refresh VisionWebDev.** Outdated tasks, prototypes, and evaluator backbones can be updated as stronger multimodal models emerge and web practices continue to change.
>
> ### 5. Presentation issues
>
> We greatly appreciate the reviewer’s helpful comments on presentation. We will improve the readability of figures in the revision and reorganize the paper structure to make the related-work discussion more natural.

---

> > ### Author Rebuttal · Reviewer_oBq1 · 2026-04-03
> >
> > My concerns have been adequately addressed

---

### Official Review · Reviewer_Brs7 · 2026-03-06

**Soundness:** 2
**Presentation:** 2
**Significance:** 3
**Originality:** 2
**Overall Recommendation:** 4
**Confidence:** 4

**Summary:**

This paper introduces VisionWebDev, a hierarchical benchmark for evaluating multimodal coding agents on visual website development. It proposes three task levels of increasing complexity (static webpage, interactive frontend, full-stack website), a structured data construction pipeline from real-world websites, and a workflow-based agent verification paradigm combining GUI agent functional testing with VLM-based visual scoring. Experiments across eight models and two frameworks reveal consistent performance degradation with task complexity, with even the best model (Claude-Opus-4.5) achieving only an average 48% on full-stack tasks.

**Compliance With Llm Reviewing Policy:**

Affirmed.

**Final Justification:**

The authors have addressed my concerns so I have raised my score to 4.

**Key Questions For Authors:**

Please refer to the weaknesses mentioned above.

**Limitations:**

The reviewer does not find an explicit discussion of the work's limitations, although the authors do mention the possible impact at the end of the paper.

**Strengths And Weaknesses:**

# Strengths

S1: **Well-motivated hierarchical design**. The three-level task formulation is a genuinely useful contribution. By disentangling static visual reproduction, interactive frontend logic, and full-stack system construction, the benchmark enables fine-grained failure attribution that monolithic benchmarks cannot provide.

S2: **Thoughtful evaluation paradigm**. The workflow-based agent verification is the paper's strongest conceptual contribution. It is likely to enable a more realistic and practical evaluation of website generation.

S3: **Good experimental coverage**. The evaluation spans eight models across two frameworks, with analysis broken down by task level, device type, website category, and functional category. The six numbered findings are clearly stated and appear to be interesting.

# Weaknesses
W1: **Limited scale at Level 3 and statistical reliability concerns.** The full-stack level contains only 27 tasks. Given the high variance in complex development tasks, this raises concerns about the statistical reliability of the reported results. For instance, differences between models in Table 3 at the full-stack level (e.g., GPT-5 at 34.1 vs. Gemini-3-Pro at 17.2 under OpenHands) may not be statistically significant with only 27 tasks. No confidence intervals or significance tests are reported.

W2: **Verifier reliability is not fully convincing.** The GUI agent verifier achieves 87.2% node-level accuracy, meaning roughly 1 in 8 verification decisions is incorrect. For a benchmark intended to provide a reliable automated evaluation, this error rate is non-trivial, especially given that errors can accumulate across multi-node workflows. The paper acknowledges this but attributes it to "model-intrinsic reasoning hallucinations" that will improve over time, a claim that is speculative. More critically, the validation sample (100 workflows from 64 tasks) may not be representative of the full benchmark. A similar issue goes to the VLM-based judge as well.

W3: **Confusing elaboration and Incomplete ablation of evaluation design choices.** The evaluation design description is confusing for the reviewer to understand, as there is no concrete example in Section 3 to clarify it. Though there is one in the Appendix, the reviewer still feels it is not clear enough for the reader to grasp this arguably most important contribution of the paper. Also, the paper does not elaborate on key design decisions in the verification paradigm. For example, how much does the guided action specification (i.e., $A_i$) in functional verification nodes actually improve reproducibility compared to objective-only specifications? What is the sensitivity of the VLM judge's visual scores to the prompt design? How do different GUI agent backbones (beyond GLM-4.6V) affect functional verification reliability?


W4: **Missing comparison with concurrent work.** The related work section mentions WebGen-Bench[1] and Vibe Bench[2] but does not provide a detailed methodological comparison. Table 1 gives only surface-level statistics. A deeper discussion of how VisionWebDev's evaluation paradigm compares to these benchmarks' approaches would strengthen the positioning. For example, the reviewer believes the Vibe benchmark also includes Agent-as-a-Verifier, which appears to be quite close to the workflow evaluation design proposed in this work.


[1] Zimu Lu, Yunqiao Yang, Houxing Ren, Haotian Hou, Han Xiao, Ke Wang, Weikang Shi, Aojun Zhou, Mingjie Zhan, Hongsheng Li. WebGen-Bench: Evaluating LLMs on Generating Interactive and Functional Websites from Scratch.

[2] MiniMax.Vibe: Visual & interactive benchmark for execution in application development.


W5: **Confusing terms.** The paper mentions a benchmark named "V-CodeBench" in Tables 6 and 7 (Appendix). Does this somehow relate to this work?

---

> ### Author Rebuttal · Authors · 2026-03-31
>
> We thank the reviewer for the constructive feedback. We address the concerns below.
>
> ### 1. Limited Level-3 scale and statistical reliability
>
> **Each Level-3 task constitutes a rich full-stack evaluation unit.** On average, a Level-3 task contains 8.5 prototypes, 28.2 test cases, and 4.3k input tokens, and typically requires 40–120 minutes to complete. The 27 tasks cover all 16 website subcategories, providing dense full-stack comparison signal while keeping annotation and evaluation cost manageable.
>
> **The hierarchy is intended to be interpreted jointly.** Level-1 and Level-2 already provide extensive evaluation signal for prerequisite capabilities underlying Level-3, such as visual reproduction and interactive frontend development. Taken together, the hierarchy supplies broad evidence across foundational and integrated abilities.
>
> **Repeated runs suggest Level-3 results are numerically stable.** We repeated the Level-3 inference and evaluation for `claude-sonnet-4-6` three times and obtained highly consistent results: VS/FS = 22.0/48.6, 21.4/47.5, and 22.8/49.3.
>
> ### 2. Verifier reliability and representativeness of validation
>
> Beyond the human-alignment results reported in the paper, we further add two sensitivity analyses showing that the reported conclusions remain stable across evaluator backbones.
>
> - **VLM judge sensitivity. The visual scores are robust at the ranking level across judge backbones.** We re-ran visual scoring with four VLM judges (gemini-3.1-pro-preview, doubao-seed-2-0-pro-260215, glm-4.6v, gpt-5.4) on 567 prototypes from 153 projects, and all four judges produced the same system-level ranking: sonnet-4-6 > gpt-5.4 > kimi-k2.5 > gemini-3.1-pro, indicating robust ranking consistency despite calibration differences.
>
> - **GUI verifier sensitivity. The GUI agent verifier is stable in score across backbones.** We compare two verifier backbones, glm-4.6v and gpt-5-mini-2025-08-07, and find nearly identical results. On Level-3 websites, the overall functional scores are 0.3136 vs. 0.3167. Both backbones preserve the same ranking and yield similar absolute scores, as shown below.
>
> |Tested model|GLM-4.6V FS|GPT-5-Mini FS|
> |-|-:|-:|
> |sonnet-4-6|48.6|46.6|
> |gpt-5.4|28.6|29.3|
> |gemini-3.1-pro-preview|17|19.1|
>
> - **Most GUI-agent inconsistencies do not materially block the verification workflow.** Residual node-level inconsistencies mainly stem from coarse validation checks or visual/structural defects that obstruct interaction, rather than arbitrary GUI-agent behavior. Under correct implementation and rendering, the GUI agent generally follows the workflow and completes the prescribed steps accurately.
>
> Together, these results show that the benchmark and the reported conclusions are robust across evaluator backbones. Additionally, we plan to update both the VLM judge and GUI agent on a quarterly basis using the latest backbone models.
>
> ### 3. Clarity of the evaluation design and missing ablations
>
> Following the reviewer’s suggestion, we will add a compact worked example in Section 3 or more directly reference the appendix example from the main text to make the evaluation design more intuitive. To further justify the design, we provide two targeted ablations:
>
> **(1) Guided-action ablation.** We evaluate the same Level-3 inference outputs from Claude Sonnet 4.6 and GPT-5.4 under two functional verification settings: the full (O, A, V) workflow and an objective-only (O, V) variant. These results indicate that **guided actions are critical for reliable multi-step functional verification.** Additional case analysis shows that, without guided actions, the GUI agent is generally unable to complete verification as intended on test cases involving 5 or more actions.
>
> |Setting|Sonnet FS|GPT FS|
> |-|-:|-:|
> |Full(O,A,V)|**48.7%**|**23.1%**|
> |Objective-only(O,V)|27.4%|20.5%|
>
> **(2) Rubric / prompt ablation for the VLM judge.** Using gemini-3.1-pro-preview, we score the same prototype set with three prompts—component-level rubric, page-level rubric, and holistic—each run three times, and report prototype-level mean variance and pairwise tie rate. The results show that **the component-level rubric best balances stability and discrimination.** Page-level scoring has lower variance, but mainly because coarse score quantization produces many ties and weak discrimination, while holistic scoring is more variable across runs.
>
> |Prompt varient|Mean std|Tie rate|
> |-|-:|-:|
> |**Component-level rubric (ours)**|4.1|**4.4%**|
> |Page-level rubric|**2.5** |26.7%|
> |Holistic prompt|5.4|13.1%|
>
> ### 4. Comparison with WebGen-Bench and Vibe Bench
>
> We outline the main distinctions from WebGen-Bench and Vibe Bench in the Introduction in terms of hierarchical task design, multimodal inputs, and workflow-based agent verification. We will strengthen this comparison and move the related-work discussion earlier for clarity. “V-CodeBench” in Appendix Tables 6 and 7 is a stale historical name and will be corrected.

---

> > ### Author Rebuttal · Reviewer_Brs7 · 2026-04-03
> >
> > Thank the authors for the detailed reply, which has basically addressed my concerns. So I have raised my score.

---

### Official Review · Reviewer_Axwd · 2026-03-12

**Soundness:** 3
**Presentation:** 3
**Significance:** 3
**Originality:** 3
**Overall Recommendation:** 4
**Confidence:** 4

**Summary:**

The paper introduces Vision WebDev, a novel hierarchical benchmark designed to evaluate the end-to-end website development capabilities of multimodal coding agents. Unlike existing benchmarks that focus on isolated code snippets or static UI-to-code tasks, Vision WebDev categorizes tasks into three progressive levels. The authors also propose a Workflow-based Agent Verification framework, which uses a GUI agent to interact with the generated site and a Vision-Language Model (VLM) to judge the output, addressing the difficulty of evaluating complex, dynamic web systems with traditional unit tests.

**Compliance With Llm Reviewing Policy:**

Affirmed.

**Final Justification:**

I am satisfied with the rebuttal and will maintain my positive evaluation. I look forward to seeing all these revisions and additional results incorporated into the final version of the manuscript.

**Key Questions For Authors:**

Reliability of the VLM Judge: While the paper mentions an 87.2% agreement with human labels, VLMs are known to have biases (e.g., favoring more detailed or aesthetically pleasing results even if functionally incorrect). A more rigorous analysis of "Evaluation Sensitivity" is needed.

Data Contamination Concerns: Since the data is based on real-world websites, there is a high probability that these sites were part of the training sets for models like GPT-4 or Claude. The authors should clarify if they performed any "de-contamination" checks or if the tasks were sufficiently modified to prevent direct memorization.

Cost and Latency: The proposed verification workflow (running a GUI agent + VLM judge) seems computationally expensive and slow compared to traditional code-based metrics. This might limit its scalability for frequent integration testing.

Failure Analysis of Level 3 Tasks: The performance at Level 3 is extremely low (Functional Score < 15%). While this highlights the difficulty, the paper would benefit from a more granular analysis: Is the failure primarily in database schema design, API routing, or the VLM's inability to follow complex multi-step instructions?

**Limitations:**

yes

**Strengths And Weaknesses:**

Strengths:

Comprehensive Hierarchical Design: The transition from static pages to full-stack development is a significant step forward in agent evaluation. It effectively mimics the real-world complexity of software engineering.

Real-world Relevance: By sourcing 193 tasks and 1,256 test cases from diverse, real-world C4 validation websites, the benchmark avoids the "toy problem" pitfall.

Innovative Evaluation Methodology: The use of "Agent-as-a-Judge" (combining GUI agents for interaction and VLMs for visual fidelity) is a clever solution to the lack of "ground truth" code in creative and functional web development.

In-depth Benchmarking: The study evaluates cutting-edge models (e.g., Claude and GPT series) across various frameworks (e.g., OpenHands), providing valuable insights into the current state of "Long-horizon" reasoning in AI agents.

Weaknesses and Questions

Reliability of the VLM Judge: While the paper mentions an 87.2% agreement with human labels, VLMs are known to have biases (e.g., favoring more detailed or aesthetically pleasing results even if functionally incorrect). A more rigorous analysis of "Evaluation Sensitivity" is needed.

Data Contamination Concerns: Since the data is based on real-world websites, there is a high probability that these sites were part of the training sets for models like GPT-4 or Claude. The authors should clarify if they performed any "de-contamination" checks or if the tasks were sufficiently modified to prevent direct memorization.

Cost and Latency: The proposed verification workflow (running a GUI agent + VLM judge) seems computationally expensive and slow compared to traditional code-based metrics. This might limit its scalability for frequent integration testing.

Failure Analysis of Level 3 Tasks: The performance at Level 3 is extremely low (Functional Score < 15%). While this highlights the difficulty, the paper would benefit from a more granular analysis: Is the failure primarily in database schema design, API routing, or the VLM's inability to follow complex multi-step instructions?

Overall, the hierarchical nature of Vision WebDev provides a roadmap for the future development of AI software engineers. Despite some concerns regarding evaluation costs and potential data leakage, the contribution is substantial and likely to be highly cited in the field of LLM-based software engineering.

---

> ### Author Rebuttal · Authors · 2026-03-31
>
> We thank the reviewer for the positive assessment of the paper. We address the key questions below.
>
> ### 1. Reliability of the VLM Judge
>
> To reduce potential visual bias, VisionWebDev uses a **rubric-guided, component-level** VLM judge instead of free-form holistic preference judgments, making the evaluation focus on **faithful reproduction** rather than aesthetic preference. Beyond the evaluator–human alignment analysis in Section 4.4, We additionally conduct a sensitivity analysis across multiple judge backbones.
>
> **Visual scoring results are consistent at the ranking level.** We re-ran visual scoring with four VLM judges (`gemini-3.1-pro-preview`, `doubao-seed-2-0-pro`, `glm-4.6v`, `gpt-5.4`) on 567 prototypes from 153 projects. All four judges produced the same system-level ranking: sonnet-4-6 > gpt-5.4 > kimi-k2.5 > gemini-3.1-pro.
>
> ### 2. Data contamination concerns
>
> **VisionWebDev is explicitly designed to reduce data contamination.** All tasks are sourced from the C4 validation split, and portions of functionality/specification logic is modified relative to the originals, so the benchmark evaluates requirement following rather than memorized retrieval. We further support this with a specification ablation on all  Level-3 tasks under three input conditions:
>
> * **Full:** complete PRD + all prototypes + resources
> * **Minimal:** short summary + homepage only
> * **No-Prototype:** short summary only
>
> | Model             | Full VS | Minimal VS | No-Prototype VS |
> | ----------------- | ------: | ---------: | --------------: |
> | claude-sonnet-4-6 |   22.0% |       3.0% |            0.8% |
> | gpt-5.4           |   16.0% |       2.3% |            0.2% |
>
> While we cannot fully rule out all possible pretraining exposure, these results suggest that **VisionWebDev primarily measures requirement following, multimodal grounding, and full-stack reasoning, rather than memorization.**
>
> - **Visual Side:** removing prototypes causes visual fidelity to collapse, rather than remain robust, suggesting that models depend strongly on the provided visual specifications instead of recalling the original websites from pretraining.
> - **Functional side:** functional scores also drop sharply on website tasks involving complex functionality (37.6 for Sonnet and 36.8 for GPT from Full to No-Prototype), indicating that end-to-end behavior depends on the provided requirements rather than stored solutions.
>
> ### 3. Cost and latency
>
> **Richer evaluation is necessary for realistic end-to-end agent assessment.** As agent benchmarks move toward long-horizon system construction, evaluation also evolves toward more realistic end-to-end assessment. Following this trend, VisionWebDev adopts a more systematic and comprehensive evaluation framework for integrated end-to-end agent performance.
>
> **In practice, the evaluation remains manageable.** Inference takes 5–20 min (Level-1), 20–40 min (Level-2), and 40–120 min (Level-3) per task; Level-3 evaluation averages about 1.5 hour per task. A full single-model run requires roughly 5–10M tokens for the VLM judge and 50–80M tokens for the GUI verifier, and can still be completed within one day.
>
> **We further improve efficiency through several engineering optimizations.** These include better reuse of prompt cache, tighter context control for the GUI agent by passing only relevant prior context instead of full screenshots histories, and dependency-aware test parallelization based on the evaluation workflow graph.
>
> ### 4. Failure analysis of Level-3 tasks
>
> Our fine-grained analysis reveals a clear capability-dependent pattern in Level-3 failures.
>
> **Weaker models mainly fail on long-horizon development itself.** Their failures often stem from incomplete implementation, omitted modules, or broken earlier-level navigation, so verification workflows are blocked before full multi-step integration can even be tested.
>
> **Stronger models mainly fail on integration and state consistency.** The dominant remaining errors are state-consistency and integration failures, especially in database persistence and API routing, with a smaller portion due to incomplete dynamic UI components.
>
> **A shared weakness across models is fine-grained requirement following.** Even when the overall system is largely built, specific details in the task specification are often only partially implemented or imprecisely followed.

---

> > ### Author Rebuttal · Reviewer_Axwd · 2026-04-03
> >
> > I would like to thank the authors for their thorough and high-quality rebuttal.
> >
> > I am satisfied with these responses and will maintain my positive evaluation. I look forward to seeing all these revisions and additional results incorporated into the final version of the manuscript.

---

### Official Review · Reviewer_1i1z · 2026-03-12

**Soundness:** 3
**Presentation:** 3
**Significance:** 3
**Originality:** 3
**Overall Recommendation:** 4
**Confidence:** 4

**Summary:**

The paper introduces VisionWebDev, a hierarchical benchmark designed to evaluate multimodal coding agents on visual website development tasks. The research aims to explore a broad topic related to the evaluation of autonomous coding agents in complex, multimodal software engineering scenarios. The paper aims to address the key issue that existing benchmarks mainly focus on limited code editing tasks or static UI generation, which do not adequately capture the full lifecycle of website development. To address this limitation, the authors construct a benchmark consisting of 193 tasks derived from real-world websites, covering three levels of complexity: static webpage generation, interactive frontend development, and full-stack website construction. Each task integrates visual prototypes, textual specifications, and multimedia resources, enabling systematic evaluation of agents’ abilities in visual understanding, cross-page reasoning, and long-horizon software development.
The benchmark also introduces a workflow-based agent verification paradigm to provide reliable and reproducible evaluation. This framework combines a GUI-based agent verifier for functional testing with a VLM-based judge that evaluates visual fidelity between generated webpages and reference prototypes. Through large-scale experiments with several state-of-the-art multimodal models and coding agent frameworks, the study reveals substantial performance gaps across different task levels and highlights persistent challenges in long-horizon planning, cross-module coordination, and multimodal reasoning. These results demonstrate the need for more robust evaluation frameworks and improved agent capabilities for complex end-to-end website development

**Compliance With Llm Reviewing Policy:**

Affirmed.

**Key Questions For Authors:**

1. The benchmark relies on a GUI agent verifier and a VLM-based judge to evaluate functionality and visual fidelity. Could the authors provide more detailed analysis on how robust these evaluators are across different website implementations and models? In particular, how sensitive are the results to the choice of the VLM judge? A stronger validation could increase confidence in the reliability of the benchmark.
2. The tasks are focused on visual website development. To what extent do the authors believe the benchmark reflects broader software engineering capabilities of coding agents beyond web development? Clarification on this point would help better understand the scope and potential impact of the benchmark.
3. The paper mentions that workflows are created with an expert-in-the-loop annotation process assisted by Claude Code. Could the authors provide more details about the annotation guidelines and quality control procedures? Additional transparency would help assess the reproducibility of the benchmark.
4. The benchmark uses a three-level hierarchical task structure. Did the authors explore alternative task decompositions or perform ablation studies to analyze how this hierarchy affects evaluation outcomes? Such analysis could help justify the design choices and strengthen the methodological contribution.

**Limitations:**

yes

**Strengths And Weaknesses:**

The paper is technically sound and presents a well-motivated benchmark design for evaluating multimodal coding agents in visual website development. The hierarchical task formulation that separates static webpages, interactive frontends, and full-stack systems provides a clear structure for analyzing model capabilities across increasing levels of complexity. The workflow-based agent verification framework, which combines a GUI agent for functional testing and a VLM-based judge for visual fidelity, is a reasonable evaluation strategy for complex end-to-end systems and helps improve reproducibility compared with loosely defined evaluation setups. The benchmark is constructed from real-world websites and includes a relatively large number of tasks, prototypes, and test cases, which increases realism and diversity. The experimental study evaluates multiple state-of-the-art models under different agent frameworks and provides detailed analyses of performance trends and failure modes. The paper is generally well organized and clearly written, and the benchmark has potential value for future research on multimodal coding agents and autonomous software development.
Despite the overall solid design, several aspects could be improved. The verification framework relies on VLM-based judging and agent execution, which may introduce additional uncertainty or bias in the evaluation process, even though the authors provide limited validation against human annotations. The benchmark mainly focuses on website development scenarios, which may limit the generality of conclusions about broader software engineering capabilities. Some implementation details of the evaluation pipeline, including workflow construction and annotation procedures, are briefly described but could benefit from more precise documentation to improve reproducibility. In addition, while the experiments compare several models and frameworks, deeper ablation studies or analysis of the benchmark design choices are limited. Finally, although the hierarchical benchmark is useful, its conceptual components build on several existing benchmarks and evaluation ideas, so the degree of methodological novelty is moderate rather than fundamentally new.

---

> ### Author Rebuttal · Authors · 2026-03-31
>
> We thank the reviewer for the positive assessment of the paper. We address the key questions below.
>
> ### 1. Evaluator robustness
>
> Evaluator reliability is particularly important in VisionWebDev. To make agent-process evaluation reliable in this setting, we adopt a workflow-based agent verification paradigm, provide structured  < O, A, V > context at each node, and use component-level visual rubrics rather than holistic judgments. We further conduct finer-grained analyses to examine the robustness of the evaluation:
>
> - **VLM judge sensitivity: The visual scores are robust at the ranking level across judge backbones.** We re-ran visual scoring with four VLM judges (gemini-3.1-pro-preview, doubao-seed-2-0-pro-260215, glm-4.6v, gpt-5.4) on 567 prototypes from 153 projects, and all four judges produced the same system-level ranking: sonnet-4-6 > gpt-5.4 > kimi-k2.5 > gemini-3.1-pro. The result indicates that visual scores are consistent in ranking despite calibration differences across judges.
> - **GUI verifier sensitivity: The GUI agent verifier is stable in score across backbones.** We compare two verifier backbones, glm-4.6v and gpt-5-mini-2025-08-07, and find nearly identical results. On Level-3 websites, the overall functional scores are 0.3136 vs. 0.3167 (absolute difference 0.003, relative difference 1.0%). The same pattern holds per model: both backbones preserve the same ranking and yield similar absolute scores, as shown below.
>
> |Tested model|GLM-4.6V FS|GPT-5-Mini FS|
> |-|-:|-:|
> |sonnet-4-6|48.6|46.6|
> |gpt-5.4|28.6|29.3|
> |gemini-3.1-pro-preview|17|19.1|
>
> Together, these results show that the benchmark is robust across evaluator backbones and the reported conclusions are not driven by single evaluator backbone. To further maintain evaluator reliability over time, we plan to update both the VLM judge and GUI agent on a quarterly basis using the latest backbone models.
>
> ### 2. Scope beyond website development
>
> **VisionWebDev is intentionally domain-focused rather than aiming for exhaustive SE coverage.** Our goal is not to cover all software engineering tasks, but to study visual, multimodal, end-to-end web development as a representative and challenging slice of SE. The benchmark captures several broadly relevant agent challenges, including long-horizon planning, modular reasoning, cross-component coordination, and multimodal grounding. Its contribution is therefore twofold:
>
> 1. a realistic benchmark for multimodal coding agents in web development
> 2. a workflow-based evaluation paradigm that may offer useful insights for building more reliable benchmarks in other SE settings.
>
> ### 3. Workflow construction and annotation transparency
>
> The workflow annotation follows a three-stage pipeline with cross-checking.
>
> **Stage 1: expert-authored workflow drafting.** Domain experts first draft high-level workflows from requirement documents and prototype designs, following principles of **decoupling dependent nodes** and **integrating related validations**. Test cases specify concrete actions and expected outcomes, and are cross-checked against requirements for coverage and traceability (e.g., “type username (example) and password (123456),” “click login,” “verify login succeeds”).
>
> **Stage 2: Claude Code-assisted workflow refinement.** Claude Code then refines these drafts into executable **objective–actions–validations** workflows while preserving requirement alignment, and the refined workflows are checked against the expert-authored drafts.
>
> **Stage 3: human quality control and verification.** Human reviewers finally examine the workflows for **completeness, correctness, consistency** with both the original requirements and the intermediate drafts.
>
> ### 4. Hierarchical design alternatives
>
> We considered several alternative decompositions during benchmark design, but each shifts the benchmark toward a different target and makes it harder to expose long-horizon failures in cross-module consistency and integration.
>
> - **By technical layer:** separating frontend, backend, database, or deployment improves failure localization, but isolates tightly coupled layers and weakens evaluation of cross-layer coordination.
> - **By product functionality:** grouping tasks by content, CRUD, authentication, e-commerce, or dashboards helps coverage analysis, but behaves more like an application taxonomy than a capability-oriented decomposition.
> - **By development stage:** decomposing tasks into planning, implementation, debugging, and refinement reflects the software lifecycle, but shifts evaluation away from whether agents can build and refine a coherent integrated system.
>
> The adopted hierarchy is designed to **preserve task continuity** while progressively **increasing integration complexity**. It follows the natural progression of web development, from visual fidelity to multi-page interactivity and full end-to-end integration. This decomposition makes the benchmark better suited to evaluating end-to-end development ability.

---

> > ### Author Rebuttal · Reviewer_1i1z · 2026-04-05
> >
> > The author's reply basically resolved my doubts.

---

### Decision · Program_Chairs · 2026-04-30

**Decision:**

Accept (spotlight)

**Comment:**

This paper presents VisionWebDev for evaluating multimodal coding agents. It has three levels of evaluation: L1 focuses on visual faithfulness of a single page, L2 focuses on interaction of the frontend between multiple pages, and L3 focuses on backend integration and persistence. This gives a full perspective of web development. The evaluation tests for functionality too.

All reviewers felt:
1. This is an important problem, and existing benchmarks focus mainly on non-visual aspects and correctness, whereas this focus on visual aspects and functionality.
2. The evaluation paradigm is thoughtful (testing for both faithfulness and functionality).
3. Strong experimental coverage.
4. Progress on this benchmark means progress on real-world tasks, as the tasks are sourced from real websites.

The rebuttal is also satisfactory, and the AC doesn't see any critical issues.